# Improving the Usefulness of Decision Trees as Explanations

## Abstract

In classification with tabular data, one often utilizes tree-based models. Those can be competitive with deep neural networks on tabular data and, under some conditions, explainable. The explainability depends on the tree's depth and the accuracy of each leaf. Decision trees containing leaves with unbalanced accuracy can provide misleading explanations. Low-accuracy leaves provide less useful explanations to the individuals they classify. Here, we train a shallow tree that minimizes the maximum misclassification error across leaf nodes. The shallow tree provides a more useful global explanation, while its overall statistical performance can approach that of state-of-the-art methods by extending the leaves with additional models, creating a partially interpretable model.

## 1 Introduction

In classification and forecasting with tabular data, one often utilizes axis-aligned decision trees (Payne & Meisel, 1977; Breiman et al., 1984). A prime example of a high-risk application of AI, where decision trees are widely used, is credit risk scoring (Mays, 1995; Lessmann et al., 2015; Thomas et al., 2017) in the financial services industry (Athey & Imbens, 2019). There, the relevant regulation, such as the Equal Credit Opportunity Act in the US (Equal Credit Opportunity Act , ECOA) and related regulation in the European Union (European Commission, 2016; 2024), is sometimes interpreted as requiring explainable models (Rudin, 2019), which is often construed as requiring the use of decision trees (Bracke et al., 2019; Dupont et al., 2020; Gunnarsson et al., 2021; Consumer Financial Protection Bureau, 2022). When studying the decision tree that a bank uses, one often focuses on ways that would make it possible to obtain a loan, and one would wish that the corresponding leaf of the decision tree had as high accuracy as possible.

In many other domains, the use of tree-based models has a long tradition. Consider, for example, judicial applications of AI such as the infamous Correctional Offender Management Profiling for Alternative Sanctions (COMPAS) (Brennan et al., 2009; Brennan & Dieterich, 2018; Courtland, 2018; Zhou et al., 2023), or medical applications of AI (e.g., Rakha et al., 2014; London, 2019; Tjoa & Guan, 2020). It is hard to overstate the importance of high accuracy of any rule that a medical doctor or a judge may learn from a decision tree. Decision trees are also used in model extraction (Bastani et al., 2017) to provide globally valid explanations of black-box classifiers.

Shallow trees can indeed serve as global explanations for a classifier—or explainable classifiers *per se*—when each leaf is construed as a logical rule. Because various individuals or groups of individuals may deem various outcomes of importance, a fair explanation tree would maximize the accuracy in each leaf of the decision tree. Additionally, the depth needs to be low[1] in order for the rule explaining the decision in each leaf to remain comprehensible.

Similarly, one could argue that a low-accuracy decision corresponds to a less useful explanation. Especially if the accuracy is near random, a small variation in the training data could lead to a different classification with the same explanation. Imagine a classification task over data containing sensitive attributes, such as race. If a discriminatory decision is found in a surrogate tree used as a global explanation of a black-box

---

[1] According to Feldman (2000), humans can understand logical rules with boolean complexity of up to 5–9, depending on their ability, where the boolean complexity is the length of the shortest Boolean formula logically equivalent to the concept, usually expressed in terms of the number of literals.

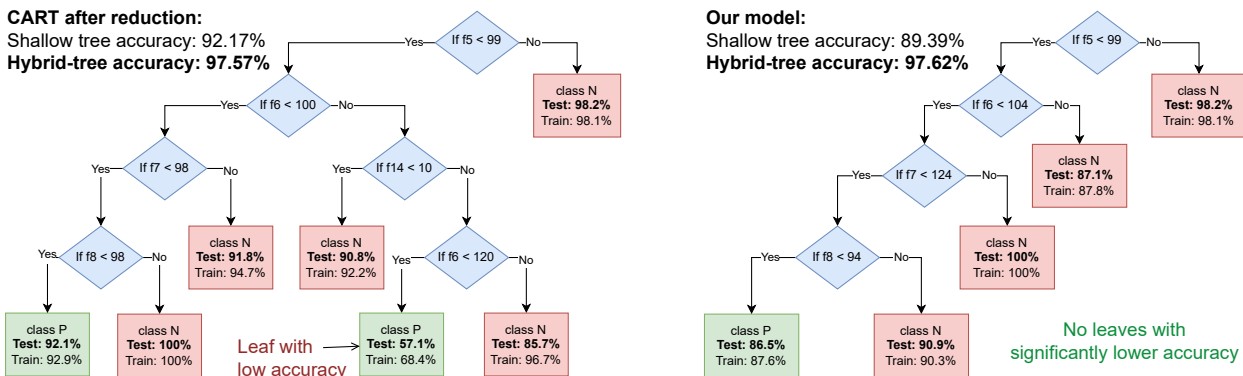

Figure 1: A comparison of decision trees produced by CART and our method for *pol* dataset. In each leaf, the bold/regular percentage shows the leaf accuracy before extending it further on the test/training data set, respectively. Below the name of the model, we present the (hybrid-tree) accuracy of the hybrid/shallow tree in bold/regular font. The CART tree contains a leaf with a notably lower accuracy compared to the overall accuracy of the model. The explanation provided by this leaf is less useful. While model accuracies do not take this into account, the proposed measure of leaf accuracy does. The left and right trees have leaf accuracy on unseen data equal 57.1% and 86.5%, respectively.

model, it may signal unfairness in that model. If, however, this leaf has low (near random) accuracy, the explanation is unreliable, since the leaf's existence might itself be a sampling artifact. Such an explanation is less useful than one that would only contain leaves with sufficiently high accuracy. Therefore, to evaluate the usefulness of explanations provided by a decision tree, we suggest considering the minimal accuracy in any leaf of the tree (tree's *leaf accuracy*).

This separation of a difficult-to-classify group from the rest of the data (thus creating a leaf with low accuracy) can be observed in trees trained with commonly used methods on real-world data. See Figure 1, which shows two trees of similar overall accuracy for the pol(e) dataset. When optimizing for overall accuracy, the minimum test accuracy in one leaf can be as low as 57.1% (cf. the left tree in Figure 1). This performance is almost equivalent to random choice, making the classification (and the explanation) dubious. However, when maximizing the minimum training accuracy in one leaf, the minimum test accuracy in one leaf increases to 86.5% (cf. the right tree in Figure 1). One could argue that this improves the usefulness of the explanation provided by the tree.

Although a recent comparison of the statistical performance of gradient-boosted trees and deep neural networks by Grinsztajn et al. (2022) has shown that the state-of-the-art tree-based models can outperform state-of-the-art neural networks across a comprehensive benchmark of tabular data sets, for our decision trees, the explainability requirement limits the overall accuracy. To create a model with comparable total accuracy, we follow the existing works that combine interpretable models with black boxes, aiming for a (tunable) balance between accuracy and explainability (e.g., Wang, 2019; Frost et al., 2024). One can extend each leaf of the tree with a separate model. Using our model as the base tree yields a more useful explanation when the extension model agrees with it. This approach suggests an analogy to PCA, where one often interprets just a few components, leaving the rest uninterpreted, using them only to improve the statistical performance. Similarly, one can train a well-balanced shallow tree that interprets the majority of the data, and the remainder is estimated by more capable models.

In our approach, we use mixed-integer optimization (MIO) to train a shallow tree, minimizing the maximum misclassification error across leaves. Put another way, we maximize the minimal accuracy in any leaf of a tree. In the second (optional) step, we can train further models that extend each leaf of the shallow tree. The shallow tree with the additional constraints on the accuracy in the leaves provides a useful explanation, while the overall accuracy of the hybrid trees (Zhou & Chen, 2002) combining shallow trees and the extending models (which we call the *hybrid-tree accuracy*) improves upon (hybrid) decision trees trained using classical

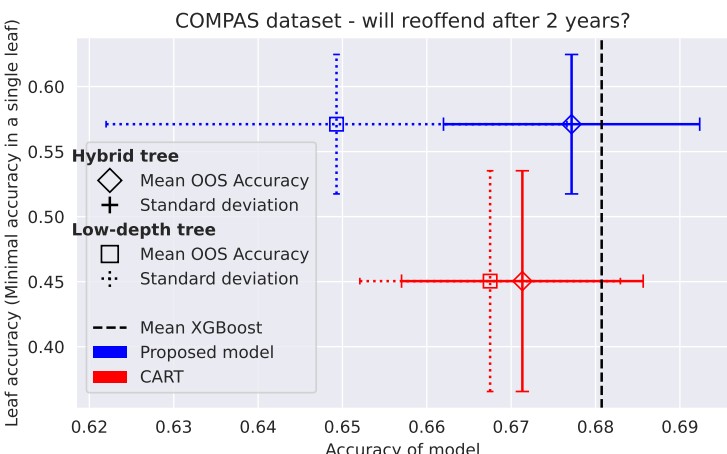

Figure 2: Performance on the COMPAS dataset: Mean statistical performance over 10 different train-test splits, evaluated in terms of model accuracy (horizontal axis) and leaf accuracy (vertical axis) for five variants of (hybrid) decision trees. The horizontal and vertical error bars are standard deviations across the 10 random runs. Notice that the proposed model has better interpretability compared to any standard decision tree and, once extended, accuracy comparable to a gradient-boosted tree.

methods (e.g., CART) and is comparable to state-of-the-art tree-based methods, such as the well-tuned XGBoost of Grinsztajn et al. (2022).

Let us illustrate the statistical performance. Figure 2 shows that the accuracy of a well-tuned XGBoost of Grinsztajn et al. on the two-year COMPAS (Brennan et al., 2009) test case exceeds 0.68. The accuracy of our shallow tree trained with the leaf-accuracy objective is below 0.65, which should not be surprising, considering the overall model accuracy is *not* the main objective. Nevertheless, by extending models in the leaf nodes of the shallow tree, we can improve the accuracy very close to 0.68, which improves upon both the accuracy of CART of the same depth alone (below 0.67) and CART of the same depth undergoing the same leaf-extension procedure (slightly above 0.67). This performance is rather typical across the benchmark of Grinsztajn et al. (2022). The proposed method outperforms CART with statistical significance, as detailed in Section 4.

**Our contributions.** We present:

- leaf accuracy as a criterion for evaluating the usefulness of a classification tree as a global explanation. Leaf accuracy of decision tree $T$ is defined as follows:

$$A_L(T) := \min_{l \in \mathcal{L}(T)} \frac{1}{|X_l|} \sum_{(\boldsymbol{x}, y) \in X_l} [\![ y = C_l ]\!] \tag{1}$$

  where $\mathcal{L}(T)$ is the set of leaf nodes of tree $T$, $X_l \subseteq X$ is the set of samples assigned to the leaf $l$, and $C_l$ is the decision class of the leaf $l$ and $[\![ a ]\!]$ equals one if $a$ is true and zero otherwise;

- a method for training decision trees that are optimal with respect to leaf accuracy;

- benchmarking on tabular datasets (Grinsztajn et al., 2022) suggesting that the leaf accuracy can be significantly improved by up to 18.41 percentage points, while suffering only a very modest drop (at most 2.76 percentage points across the benchmark) in overall model accuracy, compared to well-tuned XGBoost (Grinsztajn et al., 2022), once the shallow tree is extended.

## 2 Related Work

Decision trees are among the leading supervised machine learning methods, where interpretability and out-of-sample classification performance is important. Random forests (Breiman, 2001) and gradient-boosting tree-ensemble approaches (Mason et al., 1999; Friedman, 2001) improve upon their statistical performance substantially while limiting the interpretability. However, partially interpretable models (e.g. Wang, 2019; Frost et al., 2024) that combine an interpretable model with a black-box model have been shown to sometimes even improve the performance over pure black-box models (Le & Clarke, 2020).

We are given $n$ samples $(x_{i1}, \ldots, x_{ip}, y_i)$ with $p$ features each, for $i = 1 \ldots n$, and their classification $y_i \in [K]$ into $K$ classes. Let us denote sample $i$ by $\boldsymbol{x}_i = (x_{i1}, \ldots, x_{ip})$. A decision tree sequentially splits a set of samples into two partitions: In each non-leaf node $t$, it splits the samples based on their values $x_{\cdot j_t}$ of a particular feature $j_t \in [p]$ and a threshold $b_t$. (See Figure 1 for an illustration.) More recently, decision trees have played an important role in explainable artificial intelligence (Arrieta et al., 2020; Burkart & Huber, 2021; Souza et al., 2022) and interpretable machine learning (Rudin et al., 2022).

Construction of an optimal axis-aligned binary decision tree is NP-Hard (Laurent & Rivest, 1976), and hence all known polynomial-time algorithms, such as CART (Breiman et al., 1984), produce suboptimal results, at least for some cases. Still, CART (Breiman et al., 1984), which utilizes the Gini diversity index and cross-validation in pruning trees, ranks among the leading algorithms (Wu et al., 2008) in machine learning and is provided as the default option for training trees in popular code libraries, such as Sklearn (Pedregosa et al., 2011). A decade later, Breiman suggested that boosting can be interpreted as an optimization algorithm (Breiman, 1998), leading to the development of gradient-boosted trees (e.g., Mason et al., 1999; Friedman, 2001). Their well-tuned variants (e.g., Chen & Guestrin, 2016; Ke et al., 2017; Prokhorenkova et al., 2018) are state-of-the-art polynomial-time algorithms for training boosted trees. We refer to Gorishniy et al. (2021); Grinsztajn et al. (2022) for comparisons against deep neural networks.

Bertsimas & Dunn (2017) and, independently, others (Bessiere et al., 2009; Narodytska et al., 2018; Günlük et al., 2021), pioneered the use of exponential-time algorithms in the construction of decision trees. The MIO formulation of Bertsimas & Dunn suffers from some issues of scalability (Verwer & Zhang, 2019), but can be easily extended by the addition of further constraints, such as sparsity (Hu et al., 2019; Xin et al., 2022; Zhang et al., 2023), fairness (Verwer & Zhang, 2019; van der Linden et al., 2022), upper bounds on the number of leaves (Lin et al., 2020), incremental progress bounds (Lin et al., 2020), bounds on similarity of the support (Lin et al., 2020), a wide variety of privacy-related constraints, and in our case, accuracy in the leaves. Likewise, there are numerous extensions in terms of the objective (Lin et al., 2020), including F-score, AUC, and partial area under the ROC convex hull and, in our case, the leaf accuracy. Subsequently, the *optimal decision trees* have grown into a substantial subfield within machine learning research.

There have been several important proposals of alternative convex-optimization relaxations for optimal decision trees: Dash et al. (2018) have demonstrated the use of an extended formulation in a column-generation (branch-and-price) approach; Zhu et al. (2020) have introduced another alternative formulation and a number of valid inequalities (cuts); Aghaei et al. (2020) introduced another alternative formulation based on the maximum flow problem; Aglin et al. (2020) developed a custom branch-and-bound algorithm. Independently, Carreira-Perpinán & Tavallali (2018) suggested using non-linear optimization techniques, such as alternating minimization leading to much further research (Zantedeschi et al., 2021). In another direction, gradient-based methods have been used to train decision trees (Marton et al., 2024). Most recently, generative models were employed in creating trees (Guidotti et al., 2024), even Large Language Models directly (Knauer et al., 2025). We refer to Carrizosa et al. (2021); Nanfack et al. (2022) for overviews of mathematical optimization in the construction of decision trees, and to Costa & Pedreira (2023); Mienye & Jere (2024) for surveys of tree learning algorithms in general.

Much recent research (e.g., Vidal & Schiffer, 2020; Demirović et al., 2022; van der Linden et al., 2022; Hua et al., 2022; McTavish et al., 2022; Mazumder et al., 2022; Chaouki et al., 2024; Kohler et al., 2025) has also focussed on improving the scalability of exponential-time algorithms for optimal decision trees by using branch-and-bound methods without relaxations in the form of convex optimization and, more broadly, dynamic programming. These approaches are sometimes seen as less transparent, as the mixed-integer

formulation needs to be translated to the appropriate pruning rules or cost-to-go functions, which are less succinct, and the correctness of the translation can be non-trivial to verify. Nevertheless, Hua et al. (2022) have demonstrated the scalability of their method to a dataset with over 245,000 samples (utilizing less than 2000 core-hours), for example. This seems to validate the practical relevance of optimal decision trees.

## 3 Mixed-Integer Formulation

Mixed-Integer (Linear) Optimization (MIO) is a method of mathematical optimization similar to Linear Programming, with some of its variables limited to integer values. The goal is to maximize an objective function while satisfying a number of (linear) non-strict inequality constraints (Wolsey, 2021). Because of the global optimization capabilities, MIO enables our approach to not suffer from issues created by greedy top-down approaches like CART (e.g., Figure 1).

We build upon the MIO formulation of *optimal decision trees* (Bertsimas & Dunn, 2017), changing the objective and adding novel constraints. The entire formulation is

$$\max Q \tag{2a}$$

$$\text{s. t. } Q \leq \sum_{i=1}^{n} S_{it} + (1 - l_t) \qquad \forall t \in \mathcal{T}_L \tag{2b}$$

$$s_{it} \leq z_{it} \qquad \forall i \in [n], \ \forall t \in \mathcal{T}_L \tag{2c}$$

$$r_t \leq s_{it} + (1 - z_{it}) \qquad \forall i \in [n], \ \forall t \in \mathcal{T}_L \tag{2d}$$

$$r_t \geq s_{it} + (z_{it} - 1) \qquad \forall i \in [n], \ \forall t \in \mathcal{T}_L \tag{2e}$$

$$l_t = \sum_{i=1}^{n} s_{it} \qquad \forall t \in \mathcal{T}_L \tag{2f}$$

$$S_{it} \leq s_{it} \qquad \forall i \in [n], \ \forall t \in \mathcal{T}_L \tag{2g}$$

$$S_{it} \leq \sum_{k=1}^{K} Y_{ik} c_{kt} \qquad \forall i \in [n], \ \forall t \in \mathcal{T}_L \tag{2h}$$

$$S_{it} \geq s_{it} + \sum_{k=1}^{K} Y_{ik} c_{kt} - 1 \qquad \forall i \in [n], \ \forall t \in \mathcal{T}_L \tag{2i}$$

$$l_t = \sum_{k=1}^{K} c_{kt} \qquad \forall t \in \mathcal{T}_L \tag{2j}$$

$$\boldsymbol{a}_m^{\mathsf{T}} \boldsymbol{x}_i \geq b_m - (1 - z_{it}) \qquad \forall i \in [n], \ \forall t \in \mathcal{T}_L, \\ \forall m \in A_R(t) \tag{2k}$$

$$\boldsymbol{a}_m^{\mathsf{T}} (\boldsymbol{x}_i + \boldsymbol{\epsilon}) \leq \qquad \forall i \in [n], \ \forall t \in \mathcal{T}_L, \\ b_m + (1 + \epsilon_{\max})(1 - z_{it}) \qquad \forall m \in A_L(t) \tag{2l}$$

$$\sum_{t \in \mathcal{T}_L} z_{it} = 1 \qquad \forall i \in [n] \tag{2m}$$

$$z_{it} \leq l_t \qquad \forall i \in [n], \ \forall t \in \mathcal{T}_L \tag{2n}$$

$$\sum_{i=1}^{n} z_{it} \geq N_{\min} l_t \qquad \forall t \in \mathcal{T}_L \tag{2o}$$

$$\sum_{j=1}^{p} a_{jt} = 1 \qquad \forall t \in \mathcal{T}_B \tag{2p}$$

$$0 \leq b_t \leq 1 \qquad \forall t \in \mathcal{T}_B \tag{2q}$$

$$z_{it}, l_t \in \{0,1\} \qquad\qquad \forall i \in [n], \ \forall t \in \mathcal{T}_L \tag{2r}$$

$$a_{jt} \in \{0,1\} \qquad\qquad \forall j \in [p], \ \forall t \in \mathcal{T}_B \tag{2s}$$

$$c_{kt} \in \{0,1\} \qquad\qquad \forall k \in [K], \ \forall t \in \mathcal{T}_L \tag{2t}$$

$$0 \le Q, r_t, S_{it}, s_{it} \le 1 \qquad\qquad \forall i \in [n], \ \forall t \in \mathcal{T}_L. \tag{2u}$$

The constraints (2j – 2t) are taken from the optimal decision trees of Bertsimas & Dunn (2017), and the remaining constraints in purple from equation 2b to 2i and equation 2u, together with a different objective function equation 2a, are parts of our extensions. We use $[n]$ notation to represent the set of integers $\{1, 2, 3, \ldots, n\}$. An overview table of the variables and parameters is in the Appendix (Table 3).

### 3.1 Base model

As in the original optimal decision trees (Bertsimas & Dunn, 2017), we have $n$ samples with $p$ features each. Every point has one of $K$ classes, which is represented in the formulation by a binary matrix $Y$ such that $Y_{ik} = 1 \iff y_i = k$. All tree nodes are split into two disjoint sets, $\mathcal{T}_B$ and $\mathcal{T}_L$, which are sets of branching nodes and leaf nodes, respectively. Variable $\boldsymbol{a}_t$ is a binary vector of dimension $p$ that selects a feature $j$ to be used for decisions in node $t$. It holds that $a_{jt} = 1 \iff j$ is the selected feature in node $t$. $b_t$ is then the value of the threshold. We assume all data are normalized to the $[0, 1]$ range.

Equations (2j–2t) capture the original model of Bertsimas & Dunn (2017), wherein:

- Binary variable $c_{kt}$ is equal to 1 if and only if leaf node $t$ predicts class $k$ to data.

- Binary variable $l_t$ is equal to 1 if and only if there is any point classified by the leaf node $t$.

- Binary variable $z_{it}$ is equal to 1 if and only if point $x_i$ is classified by leaf node $t$.

The only modification to the original formulation is the omission of a binary variable $d_t$ that indicated whether a certain branching node is used. This introduced a flaw in the original formulation (Bertsimas & Dunn, 2017), which led to invalid trees, so we decided against using it. We assume it to always be 1 instead. To prune redundancies, we introduce a process of tree reduction described in Section 3.2.

Equations equation 2k and equation 2l implement the split of samples to leaf node $t$ using disjoint sets $A_R(t)$ and $A_L(t)$, containing nodes to which the leaf $t$ is on the right or on the left, respectively. Since we cannot use strict inequality, we use $\boldsymbol{\epsilon}$, a $p$-dimension vector of the smallest increments between two distinct consecutive values in every feature space (Bertsimas & Dunn, 2017):

$$\epsilon_j = \min \left\{ x_j^{(i+1)} - x_j^{(i)} \Big| x_j^{(i+1)} \ne x_j^{(i)}, \forall i \in \{1, \ldots, n-1\} \right\}$$

where $x_j^{(i)}$ is the $i$-th largest value in the $j$-th feature, $\epsilon_{\max}$ is the highest value of $\epsilon_j$ and serves as a tight big-M bound.

Finally, Equation equation 2o bounds the number of points ($N_{\min}$) in a single leaf from below.

### 3.2 MIO extensions

In the original optimal decision trees (Bertsimas & Dunn, 2017), the objective is to minimize total misclassification error. Instead, we wish to maximize the leaf accuracy. Because a single sample usually contributes differently to accuracy at different leaves, we need to introduce multiple new variables to track the accuracy in each leaf:

- variable $s_{it}$ represents the potential accuracy that sample $\boldsymbol{x}_i$ has in leaf $t$. It takes values in the range $[0, 1]$ and must sum to 1 when summing across all samples assigned to leaf $t$. This is ensured by setting the value to 0 for all points that are not assigned to the leaf $t$ in constraint equation 2c. The sum of 1 is enforced in constraint equation 2f for non-empty leaves. Empty leaves do not have non-zero $s_{it}$ values for any $i$ and thus could not sum to 1.

- reference accuracy variable $r_t$ serves as a common variable to which all accuracy contributions are equal. This is, of course, required only for points assigned to the leaf $t$. This is enforced in equation 2d and equation 2e.

- variable $S_{it}$ represents the true assignment of accuracy given by the sample. That is achieved by setting it to 0 for misclassified points using constraint equation 2h and by setting it equal to $s_{it}$ otherwise by constraints equation 2g and equation 2i.

- variable $Q$ is our objective and represents the *leaf accuracy* of the tree. Equivalent to $A_L(T)$ defined in Equation equation 1, it is the lowest achieved accuracy across all non-empty leaves as per constraint equation 2b. For empty leaves, this constraint will be trivially satisfied since $Q$ cannot take a value higher than 1 anyway.

**Tree reduction**    After the optimized tree is recovered from the formulation, empty leaves are pruned to obtain the resulting unbalanced tree. Furthermore, to account for suboptimal solutions obtained when the solver is run with a strict time limit, each pair of sibling leaves classified in the same class is merged. This is performed recursively until no further action can be performed. This leads to no loss in model accuracy and, in many cases, to an improvement in leaf accuracy, since we consider the minimum over the leaves, which cannot decrease when combining two leaves with the same majority class into one. The effect of tree reduction is shown in Figure 9.

**Tree extension**    Optionally, we can extend the leaves with further models to improve the full model accuracy (hybrid-tree accuracy). In experiments, we used XGBoost as the extension model since it was the best-performing model on the used benchmark (Grinsztajn et al., 2022). We trained a separate model for each leaf of the shallow tree after the aforementioned reduction. The models' hyperparameters were tuned using 50 iterations of a Bayesian hyperparameter search with 3-fold cross-validation at each leaf. In experiments, we reduce and extend trees generated by other methods (OCT, CART) in the same way.

## 4   Numerical results

We have implemented the method in Python, and all code and results are provided in the Supplementary material. We will release them under an open-source license to GitHub once the paper has been accepted. The hyperparameters have been chosen as follows:

- The shallow trees have been trained using the formulation in equation 2 with depth limited to four since that is a reasonable threshold for interpretability (e.g., printability on an A4 page, similar to Figure 1) and for not diluting the dataset to small parts that would impede the ability to train the extending models.

- To further support this, we set the minimal amount of points in a leaf ($N_{\min}$) to 50.

- MIPFocus and Heuristics hyperparameters were set to 1 and 0.8, respectively, to focus on finding feasible solutions in the search since that leads to the fastest improvements of the solution. However, our experiments in Appendix A.3.1 show that default MIO solver hyperparameters perform similarly.

We performed our experiments on the benchmark of Grinsztajn et al. (2022), shared under a permissive CC-BY license via the OpenML platform (Vanschoren et al., 2013). The benchmark contains medium-sized real datasets for both regression and classification of tabular data. Tree-based models are the best performing on these datasets (Grinsztajn et al., 2022), making the datasets fitting for our purpose. Since our implementation considers only classification, we consider only classification datasets. Grinsztajn et al. (2022) divide the datasets into numerical datasets and datasets with some categorical features. We follow this distinction and present results on both kinds of datasets separately. We also follow the suggestion of Grinsztajn et al. to perform 10 different train-test splits with at most 10,000 data points or 80% of total data points (whichever is lower) for training across all datasets. That is, each model has been trained on each dataset 10 times, with different seeds for data splits. The training used 80% of all data points or 10,000 data

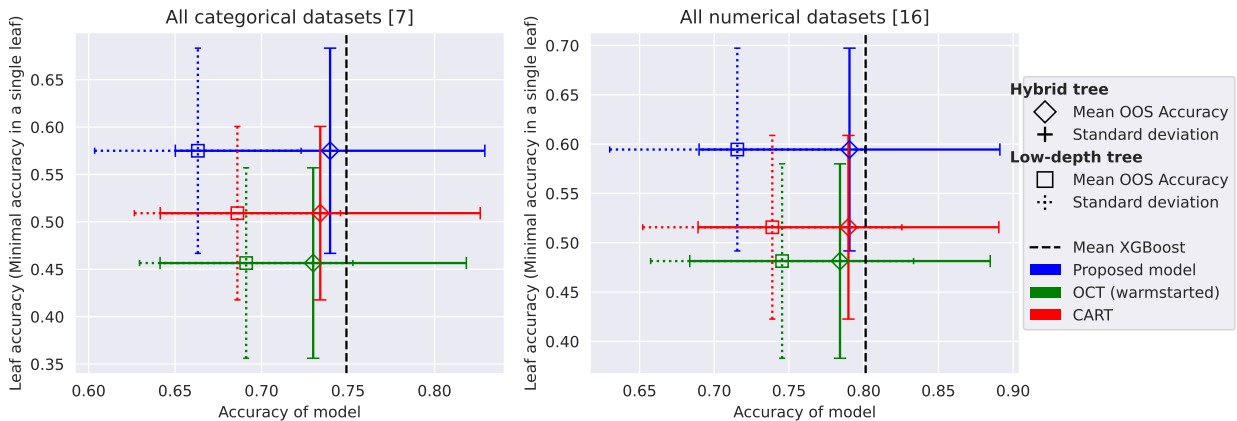

Figure 3: Results on out-of-sample data. The plot shows a significant increase in leaf accuracy when using our method, significantly improving the usefulness of the explanations provided. It also shows an increase in model accuracy when extending the models with XGBoost models in leaves. The results of the OCT model serve to compare to the model we built upon. We measure leaf accuracy of the shallow trees only, before extensions.

points, whichever is lower, while the remaining 20% of the dataset has been used as the test set for evaluating the model accuracy and leaf accuracy $A_L(T)$—see Equation 1. The MIO of Proposed method and OCT were warmstarted using a CART solution trained on the same data with default scikit-learn parameters, except for maximal depth and a minimal number of samples in a leaf, which were set to 4 and 50, respectively.

We performed all experiments on an internal cluster with sufficient amounts of memory. Each run of the MIO solver has been limited to 8 hours on 8 cores of an AMD Epyc 7543, totaling 64 core-hours per dataset split. The extension part takes, on average, about 3 core hours per split. This totals around 15,500 core-hours for the entire classification part of the tabular benchmark and one configuration of hyperparameters.

We compare our method of training classification trees to CART, as it is by far the most common. All experiments used the scikit-learn implementation of CART, also utilizing the option of cost complexity pruning. The hyperparameters for CART were optimized using Bayesian hyperparameter optimization for 100 iterations using 5-fold cross-validation. Hyperparameter search space was notably constrained only by fixing a maximal depth to 4 and a minimal number of samples in leaves to 50, ensuring comparability to our shallow trees. In comparison to unconstrained depth CART and CART with optimized lower bound on the number of samples in a leaf, our model interestingly fared even better. See Appendix A.12.1 for details. The entire optimization of CART with the extensions of the leaves took around 500 core-hours for the entire benchmark. The XGBoost results are taken from the authors of the paper introducing the benchmark, which suggests 20,000 core-hours have been spent producing these. (Grinsztajn et al., 2022)

Figure 3 shows the average performance (model accuracy and leaf accuracy) over categorical and numerical datasets. We include the comparison to optimal classification trees (OCT) (Bertsimas & Dunn, 2017) since it is the formulation we built on. The OCT model is warmstarted the same way as the proposed model and has the worst performance. The proposed model improves the leaf accuracy by 7.49 percentage points on average compared to CART and by 11.47 percentage points compared to OCT. The large error bars in Figure 3 might suggest that the performance varies, but this spread is largely influenced by the varying difficulty of the datasets. We thus present normalized leaf accuracy (model's leaf accuracy divided by the leaf accuracy of a model with the highest leaf accuracy on a given data split) in Figure 4. The proposed method achieves the best performance for more than 80% of the cases, while both other methods achieve it in less than 15%.

Table 1 quantifies the differences numerically. Our partially interpretable model has worse accuracy by about 1 percentage point on average when compared to the uninterpretable, best-performing state-of-the-art model (XGBoost). Our approach improves the accuracy of hybrid-tree models built on CART trees. But more importantly, it improves the leaf accuracy.

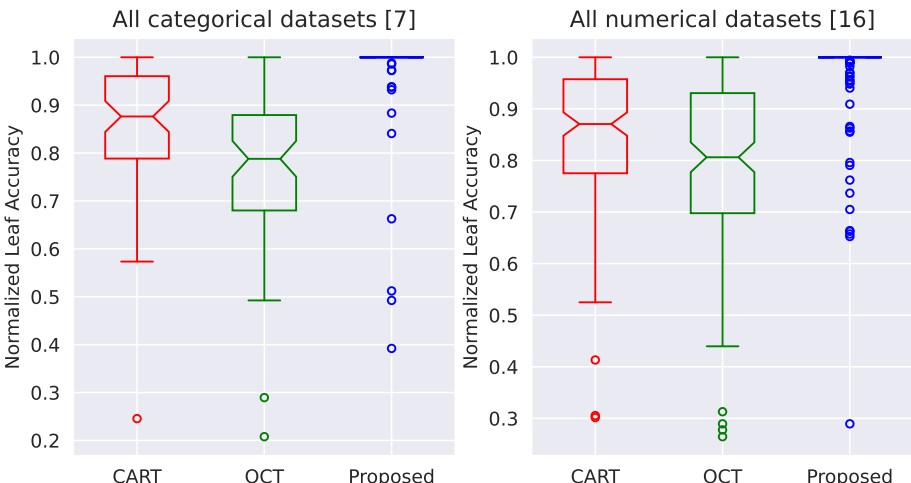

Figure 4: Comparison of normalized leaf accuracy (on test data) of the 3 methods. The boxes cover the range from the first to the third quartile. Proposed method scores best for over 80% of the runs, collapsing the box.

Table 1: Improvements in mean accuracy on datasets between our model and comparable models. Data is computed by subtracting the mean accuracy of CART or XGBoost, respectively, on each dataset from the mean accuracy of our model. In the first two rows, we compare the leaf accuracy of our shallow model to CART. In the middle two rows, we compare the hybrid-tree accuracy of the extended trees with CART trees extended in the same way. In the last two rows, we compare our extended hybrid-tree model to the mean XGBoost model trained on the same dataset. Visual comparison, including the OCT model, is in Figure 3.

|  | Baseline | Data type | Minimum | Mean ($\pm$ std) | Maximum |
|---|---|---|---|---|---|
| **Leaf Accuracy** | CART | categorical | $-0.0142$ | $0.0569 \pm 0.0533$ | $0.1206$ |
|  |  | numerical | $-0.0061$ | $0.0770 \pm 0.0556$ | $0.1841$ |
| **Hybrid-tree Accuracy** | CART | categorical | $-0.0078$ | $0.0040 \pm 0.0071$ | $0.0147$ |
|  |  | numerical | $-0.0244$ | $0.0004 \pm 0.0082$ | $0.0087$ |
| **Hybrid-tree Accuracy** | XGBoost | categorical | $-0.0228$ | $-0.0095 \pm 0.0064$ | $-0.0036$ |
|  |  | numerical | $-0.0276$ | $-0.0108 \pm 0.0076$ | $0.0005$ |

**Scalability**  MIO is known to require substantial computational effort. In our experiments, training requires between 15 and 95 GB of working memory. Details are provided in the Appendix (Figure 7). In our setting, the Gurobi solver closes the MIP Gap to around 60% on average after eight hours, as shown in Figure 5b. A 60% MIP gap means the global optimum lies between the incumbent solution and its 1.6 times multiple. In general, however, the solver can spend a disproportionate amount of time closing the gap while the optimal solution is found early on in the process. In our experiments, even though the MIP gap is rather wide and we do not certify optimality, the proposed model outperforms CART even after just 1 hour of computation; see Figure 8c in the Appendix.

**Statistical significance**  Demšar (2006) summarizes statistical tests used for the comparison of algorithms on multiple datasets. Compared to CART, the proposed method has better leaf accuracy and better hybrid-tree accuracy on a substantial majority of datasets. Using the basic sign test (Demšar, 2006), both results are statistically significant with $p < 0.05$. Using the Wilcoxon signed-rank test, we reject the null hypothesis that CART performs better than the proposed method with high confidence ($\alpha \leq 0.01$ for leaf accuracy and $\alpha \leq 0.05$ for hybrid-tree accuracy). For further results, refer to Appendix A.7.

## 5 Conclusion

We have identified an important problem of the *usefulness* of a tree as an explanation. We have shown that tree learning methods (both greedy heuristic and globally optimizing) do leave room for improvement in terms of leaf accuracy. Our approach offers multiple benefits.

First, it ensures better usefulness of every explanation provided, improving the leaf accuracy by around 7 percentage points on average across the benchmark of tabular datasets (Grinsztajn et al., 2022).

Second, if the generated trees are extended into hybrid trees, the model accuracy improves over that of similarly extended trees constructed using integer optimization, as well as hybrid trees in which the base shallow tree is obtained using CART. This suggests that using leaf accuracy might be beneficial in creating the base trees in hybrid-tree models. We leave examining this research question for future work.

Finally, the formulation is easily extensible to include additional constraints, such as shape constraints. Overall, we hope that the proposed approach may improve the usefulness of decision trees as explanations.

**Limitations**  Similar to most partially interpretable models, the hybrid-tree model aims to strike a balance between global explainability (a pure tree) and model accuracy (a black-box XGBoost). The extension by black-box models limits the use of the shallow tree as an explanation, especially in cases when the extending model changes the decision of the shallow tree. In our experiments, the shallow tree agrees with the extending models in 81.3% of the cases on average across all the datasets tested. This approach thus allows the choice between a explainable shallow tree with lower accuracy or a highly accurate model that provides a simple explanation for a majority of the cases. This "agreement rate" for the proposed method is slightly lower than that of CART trees, which is understandable given the higher accuracy of the shallow tree. For more discussion of agreement rates, see Appendix A.11. By using a more explainable model than XGBoost to extend the leaves, one could provide explanations even for the remaining classifications.

In the deployment of hybrid models, one must be cautious not to mislead users into believing that the shallow tree captures the model's decision entirely. The disagreement rate should be computed, and a model should be used only when disagreement is low, to avoid misleading the users. The use of shallow trees without extensions is always the most transparent (fully interpretable) choice and should be considered where appropriate.

Here, we only touch on the topic of hybrid trees, since the main goal of this work is the examination of leaf accuracy. Future work should focus in greater depth on the effects of training trees with high leaf accuracy for use in hybrid trees. Our results suggest that high leaf-level accuracy in the base model might improve overall model performance.

The proposed approach shares some of the scalability limitations of the original optimal decision trees (Bertsimas & Dunn, 2017). On the other hand, our method achieves good performance on real-world datasets. Furthermore, recently proposed methods (e.g., Vidal & Schiffer, 2020; Demirović et al., 2022; van der Linden et al., 2022; Hua et al., 2022; Mazumder et al., 2022; Boutilier et al., 2022) improving the scalability of optimal decision trees can be applied, in principle.

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

## A Appendix

In the Appendix, we provide the source code, complete results in `.csv` files, and a Jupyter notebook with example tests. All will be publicly available once the paper is accepted. Here, we present the results of further tests performed, describe ablation analyses, and provide further details about the results already presented.

Table 2: Listed classification datasets of the tabular benchmark. Train sets contained 80% of the total amount of samples truncated to at most 10 000 samples. 16 datasets affected by this have their number of samples in bold.

| categorical datasets | # of samples | # of features | # of classes |
|---|---|---|---|
| albert | **58252** | 31 | 2 |
| compas-two-years | 4966 | 11 | 2 |
| covertype | **423680** | 54 | 2 |
| default-of-credit-card-clients | **13272** | 21 | 2 |
| electricity | **38474** | 8 | 2 |
| eye_movements | 7608 | 23 | 2 |
| road-safety | **111762** | 32 | 2 |
| **numerical datasets** | **# of samples** | **# of features** | **# of classes** |
| bank-marketing | 10578 | 7 | 2 |
| Bioresponse | 3434 | 419 | 2 |
| california | **20634** | 8 | 2 |
| covertype | **566602** | 32 | 2 |
| credit | **16714** | 10 | 2 |
| default-of-credit-card-clients | **13272** | 20 | 2 |
| Diabetes130US | **71090** | 7 | 2 |
| electricity | **38474** | 7 | 2 |
| eye_movements | 7608 | 20 | 2 |
| Higgs | **940160** | 24 | 2 |
| heloc | 10000 | 22 | 2 |
| house_16H | **13488** | 16 | 2 |
| jannis | **57580** | 54 | 2 |
| MagicTelescope | **13376** | 10 | 2 |
| MiniBooNE | **72998** | 50 | 2 |
| pol | 10082 | 26 | 2 |

## A.1 Datasets

We used the classification part of the data sets from the mid-sized tabular data put together by Grinsztajn et al. (2022). The datasets, with their properties, are listed in Table 2. Training sets contained 80% of the total amount of samples truncated to at most 10,000 samples. This constraint affects 16 of the 23 total datasets, although some only marginally. The affected datasets have their number of samples in Table 2 in bold. The remaining 20% of the samples were the testing dataset. We used 10 random seeds that determined the train-test splits of each dataset and fixed the randomness in the training process. The seeds were namely integers 0 to 9.

Additionally, datasets are either categorical or numerical. Categorical are those that contain at least one categorical feature. Numerical datasets have no categorical features. Four numerical datasets are the same as categorical datasets but with their categorical features removed (`covertype`, `default-of-credit-card-clients`, `electricity`, `eye_movements`). Only datasets without missing features and with sufficient complexity are included in the benchmark. For more details on the methodology of dataset selection, we refer to the original paper (Grinsztajn et al., 2022).

## A.2 MIO formulation description

We provide Table 3 with short descriptions of the parameters and variables in the MIO formulation of the proposed model from equation 2.

Table 3: Description of MIO symbols used in the proposed formulation in equation 2. Parameter $n$ refers to the number of samples, $K$ is the number of classes, $p$ is the number of features, and $d$ is the depth of the tree.

|  | Symbol | Explanation | Size |
|---|---|---|---|
| Params | $Y_{ik}$ | Equal 1 for true class of a sample | $n \times K$ |
|  | $\boldsymbol{x}_i$ | Input samples | $n \times p$ |
|  | $\boldsymbol{\epsilon}$ | Minimal change in feature values | $p$ |
|  | $\epsilon_{\max}$ | Maximal value of $\boldsymbol{\epsilon}$ | 1 |
|  | $N_{\min}$ | Minimum of samples in a leaf | 1 |
|  | $\mathcal{T}_L$ | Set of leaf nodes | $2^d$ |
|  | $\mathcal{T}_B$ | Set of decision (branching) nodes | $2^d - 1$ |
|  | $A_L(t)$ | Ancestors of leaf $t$ that decide left | $\leq d - 1$ |
|  | $A_R(t)$ | Ancestors of leaf $t$ that decide right | $\leq d - 1$ |
| Variables | $Q$ | Tree's leaf accuracy | 1 |
|  | $s_{it}$ | Accuracy potential of $\boldsymbol{x}_i$ in leaf $t$ | $n \times 2^d$ |
|  | $S_{it}$ | Accuracy contribution of $\boldsymbol{x}_i$ in leaf $t$ | $n \times 2^d$ |
|  | $r_t$ | Reference accuracy for $s_{:t}$ | $2^d$ |
|  | $z_{it}$ | Assignment of $\boldsymbol{x}_i$ to leaf $t$ | $n \times 2^d$ |
|  | $l_t$ | Non-emptiness of leaf $t$ | $2^d$ |
|  | $c_{kt}$ | Assignment of class $k$ to leaf $t$ | $K \times 2^d$ |
|  | $a_{jt}$ | 1 if deciding on feature $j$ in node $t$ | $p \times (2^d - 1)$ |
|  | $b_t$ | Decision threshold in node $t$ | $2^d - 1$ |

### A.3 MIO Solver

We have utilized the Gurobi optimizer as an MIO solver under an academic license. Although the solver makes steady progress towards global optimality, the road there is lengthy. Figure 5b shows the progress of the MIP Gaps during the 8-hour optimization averaged over all datasets. For a detailed, per-dataset view, see Figure 6. The solution is still improving, albeit rather slowly, after 8 hours. The narrowing of the MIP gap is achieved only by finding better feasible solutions. This lack of improvement of the objective bound might have been affected by our hyperparameter settings which focused on finding feasible solutions and heuristic search. However, tests with default parameters did not improve the best bound either.

### A.3.1 Default hyperparameters of Gurobi solver

The performance of the Gurobi optimizer depends on the choice of hyperparameters. For the sake of simplicity, we have considered only two sets of parameters. To measure the performance change of our choice of (hyper)parameters, we ran a test with the default value of the MIPFocus parameter and a test with the default value of the Heuristics parameter.

The results (cf. Figure 5a and Tables 4, 5) show no significant improvements regarding the MIOFocus parameter. However, with the default value of the Heuristics parameter, we observe an improvement in performance on numerical datasets and a decrease in performance on categorical datasets. Both absolute differences in accuracy are about 0.015, so we opted for the variant with similar performances on both categorical and numerical datasets. That is the proposed variant focusing on heuristics. This proposed configuration also shows a more stable increase in accuracy w.r.t. the performance of CART models. The solver performance varies per dataset, as visualized in Figure 6.

These differences in performance suggest that hyperparameter space regarding the MIO solver should be further explored and could yield improvements. A closer look at Figure 6 suggests that different configurations help achieve better conditions for the solver on different datasets. This might be an area of further hyperparameter tuning based on the specific attributes of the dataset.

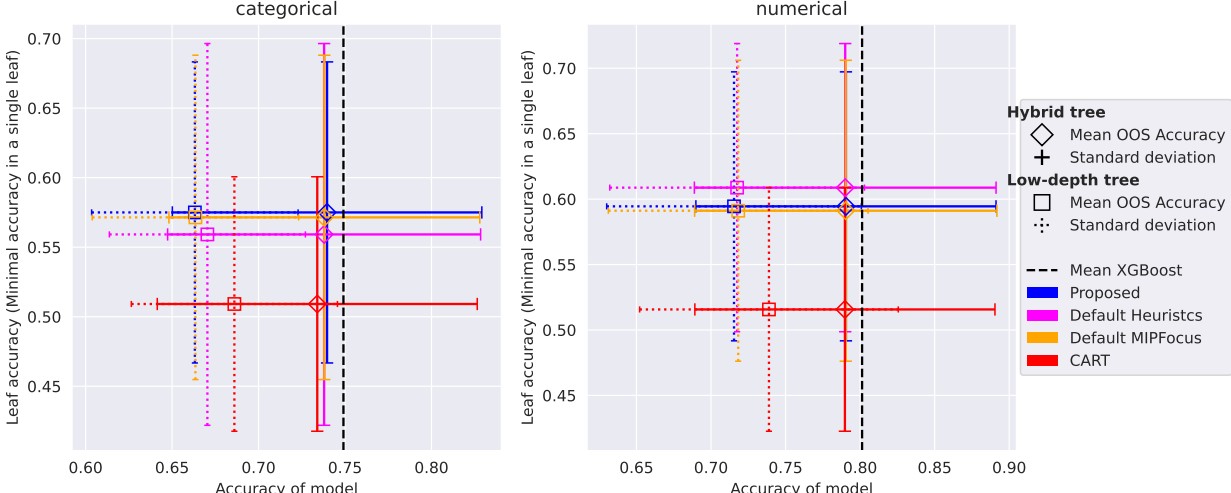

(a) Comparison of the Proposed model to models with default parameter configurations shows varying results. MIP-Focus seems to influence the search only very slightly. Heuristics, on the other hand, show significant improvement on numerical datasets and a decrease in performance on categorical datasets, with about the same absolute difference.

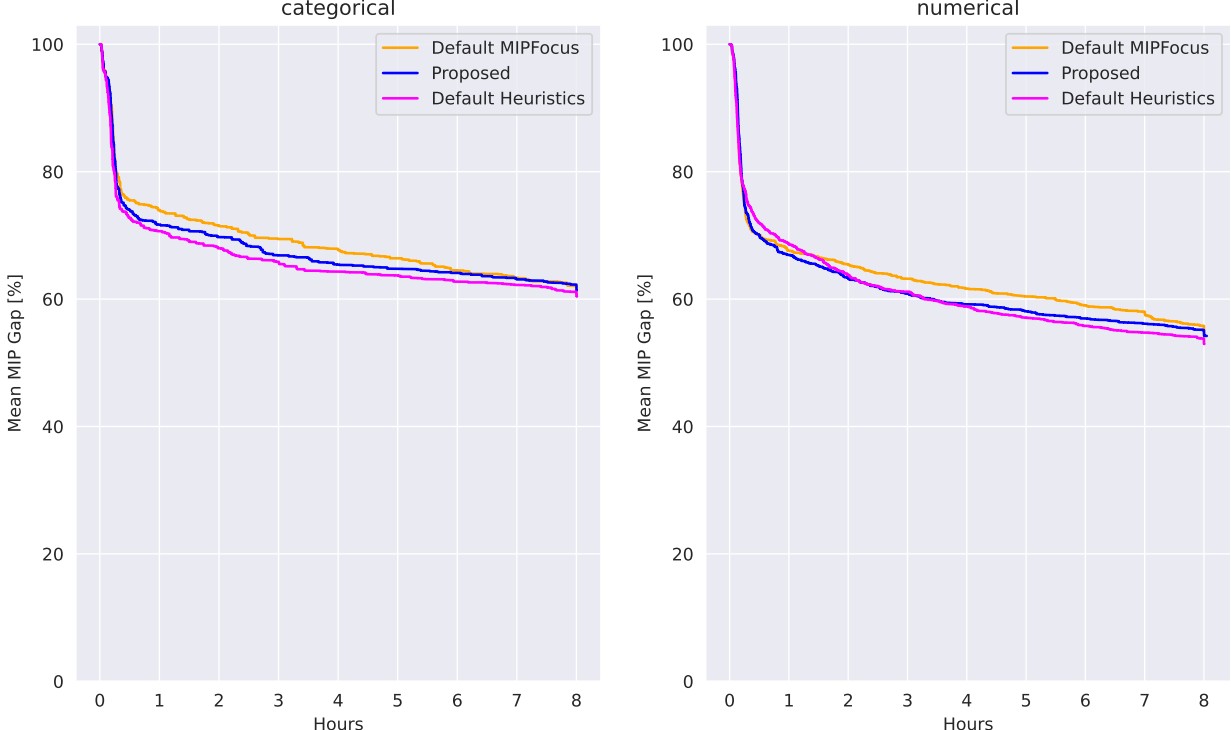

(b) Mean MIO optimality gap development over the solving time, averaged over all datasets. For a non-aggregated version, see Figure 6.

Figure 5: Comparison of models with the proposed configuration of Gurobi hyperparameters and runs with default values of the modified parameters.

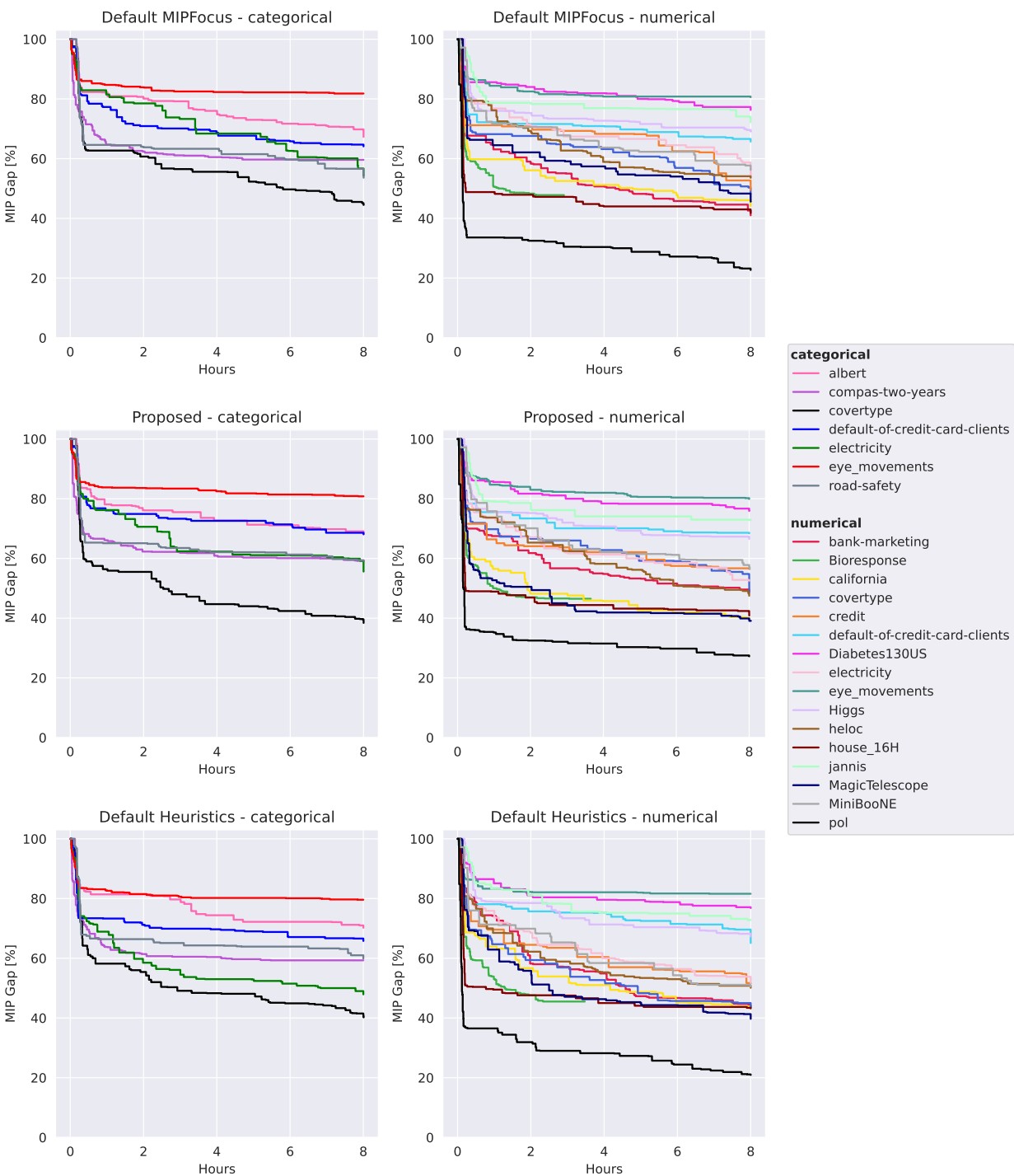

Figure 6: Mean MIO optimality gap development over the solving time averaged over 10 different train-test splits. The figure shows the progress of the value of the MIO optimality gap averaged over all splits of each dataset. Each line corresponds to one dataset. For an aggregated version, see Figure 5b.

Table 4: Detailed view of the differences in the accuracy between the default Heuristics parameter and the proposed configuration (Heuristics = 0.8). A positive number means the accuracy advantage of the proposed hyperparameter configuration. We see absolute mean differences of comparable values. The negative difference in leaf accuracy on numerical datasets also has a higher standard deviation, suggesting a stronger influence by an outlier dataset. For a graphical representation of this data, see Figure 5a.

|  | Data type | Minimal | Mean ($\pm$ std) | Maximal |
|---|---|---|---|---|
| **Leaf Accuracy** | categorical | $-0.0117$ | $0.0158 \pm 0.0234$ | $0.0531$ |
|  | numerical | $-0.1178$ | $-0.0143 \pm 0.0402$ | $0.0435$ |
| **Hybrid-tree Accuracy** | categorical | $-0.0011$ | $0.0017 \pm 0.0035$ | $0.0094$ |
|  | numerical | $-0.0047$ | $0.0005 \pm 0.0025$ | $0.0062$ |

Table 5: Detailed view of the differences in the accuracy between the default MIPFocus parameter and the proposed configuration (MIPFocus = 1). A positive number means the accuracy advantage of the proposed hyperparameter configuration. Both variants seem to perform comparably, with a potential slight edge in favor of the proposed configuration. For a graphical representation of this data, see Figure 5a.

|  | Data type | Minimal | Mean ($\pm$ std) | Maximal |
|---|---|---|---|---|
| **Leaf Accuracy** | categorical | $-0.0304$ | $0.0036 \pm 0.0213$ | $0.0299$ |
|  | numerical | $-0.0528$ | $0.0034 \pm 0.0342$ | $0.0788$ |
| **Hybrid-tree Accuracy** | categorical | $-0.0028$ | $0.0016 \pm 0.0043$ | $0.0088$ |
|  | numerical | $-0.0026$ | $0.0001 \pm 0.0019$ | $0.0032$ |

## A.4 Memory requirements

Overall, the memory requirements of the datasets were between 15 and 95 GB. On average, all datasets required at most 70 GB of working memory. Figure 7 shows the memory requirements of our formulation in more detail. The extension phase of the process is negligible in this regard, as it requires only about 1.5 GB of working memory in total and is performed after the MIO optimization. Training and extending the CART models also required less than 2 GB of working memory.

The amount of memory required by the MIO solver is dependent on the size of the data in the number of training samples, as well as the number of features. Figure 7b shows this linear dependence of memory requirements on the size of the training set. Based on the coloring of the nodes, we also see the dependence on the number of features, especially in the case of the Bioresponse dataset.

### A.4.1 Performance of the model given a shorter time

When considering a shorter time for optimization, we can lower the memory requirements to levels attainable by current personal computers. When optimizing our MIO model for one hour, the required memory is below 50 GB for all datasets except Bioresponse, which has one order of magnitude more features than the rest of the datasets included in the benchmark. The mean memory requirement is below 30 GB of working memory (compared to 50 GB for the 8-hour run). See Figure 8 for details.

Figure 8c shows that even with this limited budget, we can achieve significant improvement compared to CART in leaf accuracy and similar accuracy of hybrid trees.

## A.5 Reduction of the trees

The reduction phase has a beneficial influence on the leaf accuracy of a model. Figure 9 shows this improvement in mean leaf accuracy over all datasets.

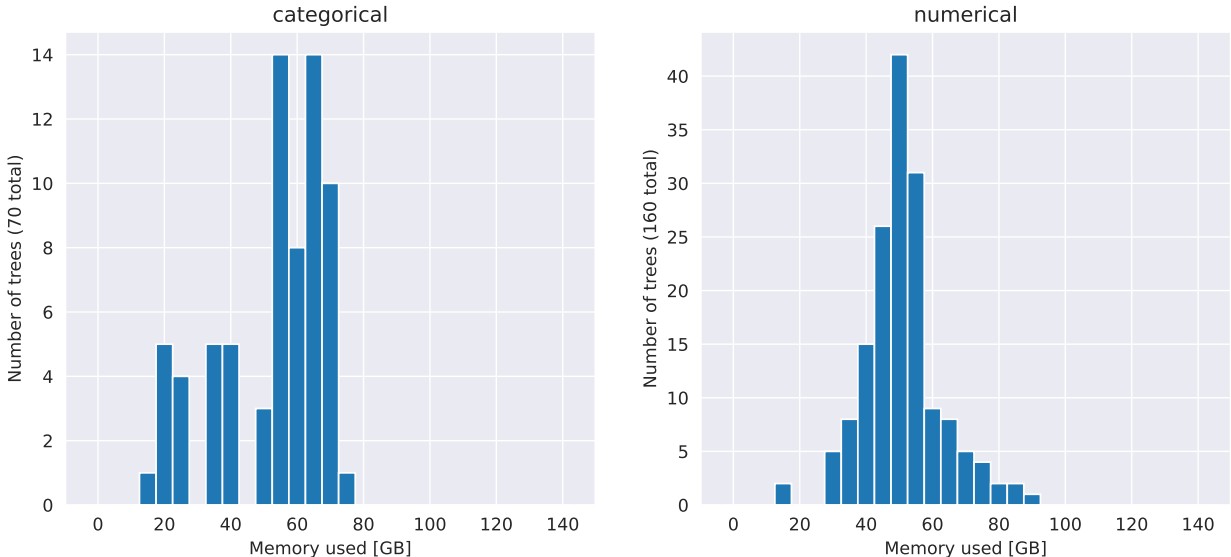

(a) Histogram of memory requirements of MIO solver for all dataset splits.

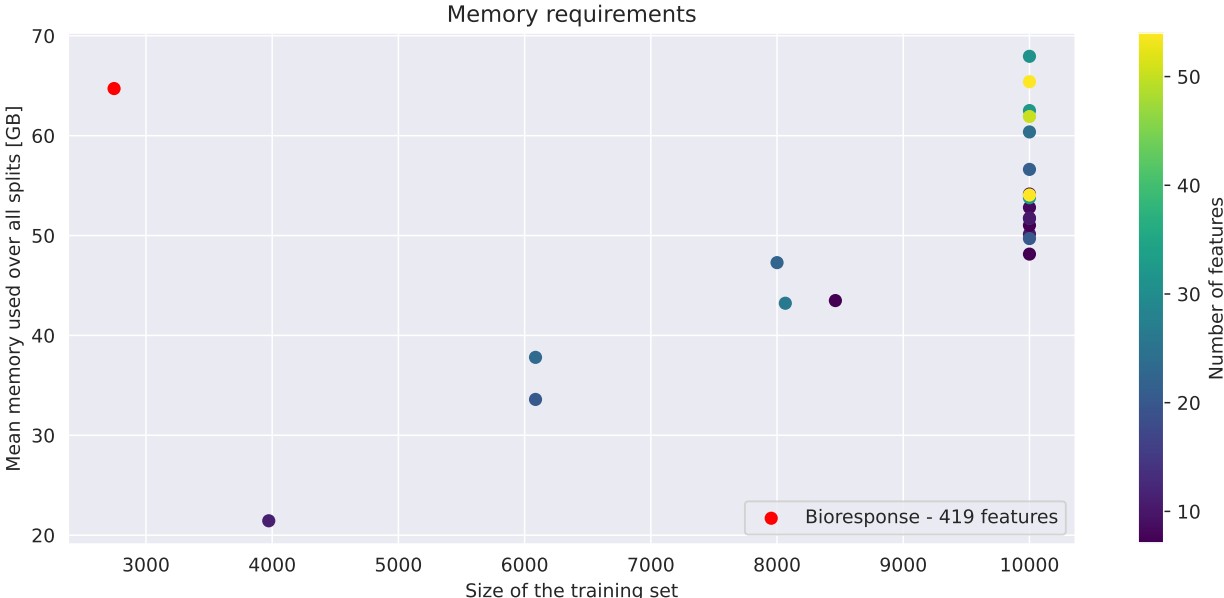

(b) Mean memory requirements on datasets. Dots are colored according to the number of features. Dataset Bioresponse is excluded from the color mapping due to having significantly more features. Training sets were truncated to a maximum of 10,000 points.

Figure 7: Memory requirements mostly do not exceed 70 GB. The memory requirements increase slightly when more time is given to the solver and significantly increase when bigger training sets are considered. We can also see some correlation between the number of features and memory requirements when looking at same-size datasets.

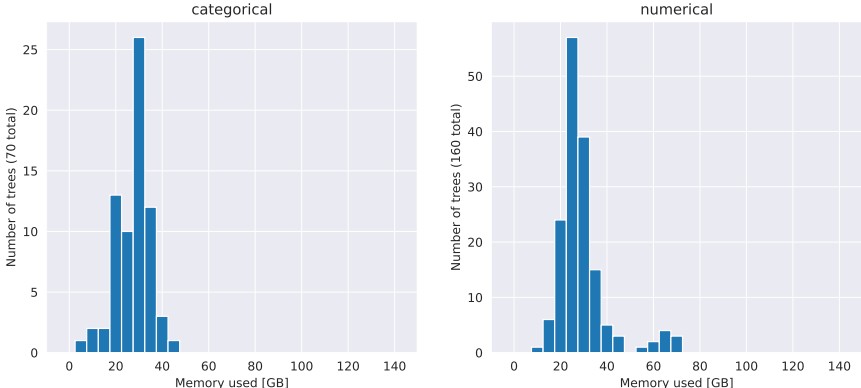

(a) Histogram of memory requirements of MIO solver for all dataset splits.

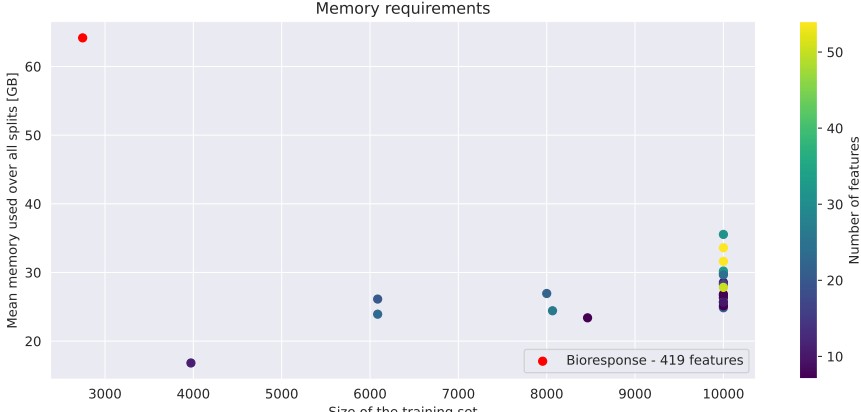

(b) Mean memory requirements on datasets. Dots are colored according to the number of features. Dataset Bioresponse is excluded from the color mapping due to having a significantly higher number of features. Training sets were clipped to a maximum of 10,000 points.

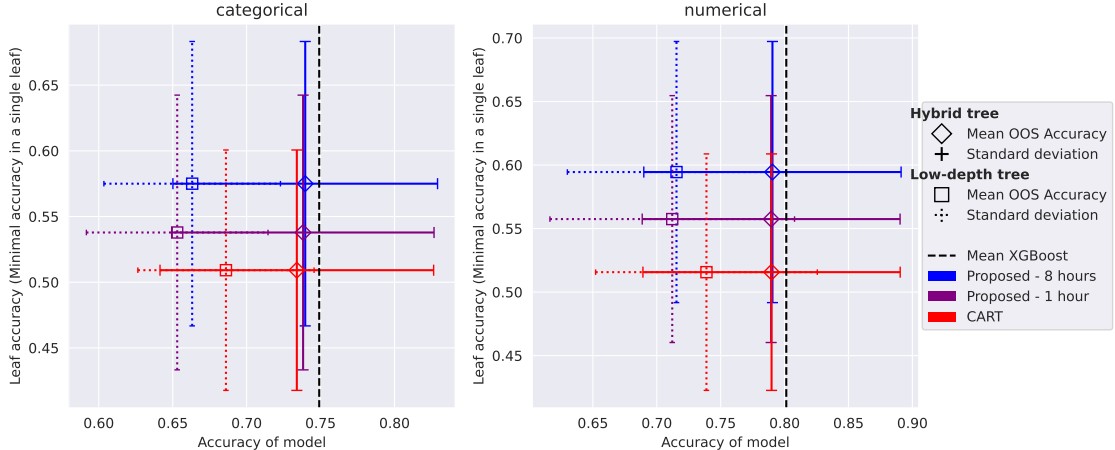

(c) Comparison of the performance of the Proposed model after 1 and 8 hours of optimization.

Figure 8: Comparison to a version of the Proposed model that the Gurobi solver optimized for only one hour. Compared to the main configuration, which ran for 8 hours, we notice a significant decrease in memory requirements for most datasets, up to tens of gigabytes. An outlier dataset Bioresponse with cca 10 times more features sees a smaller decrease of about 2 GB.

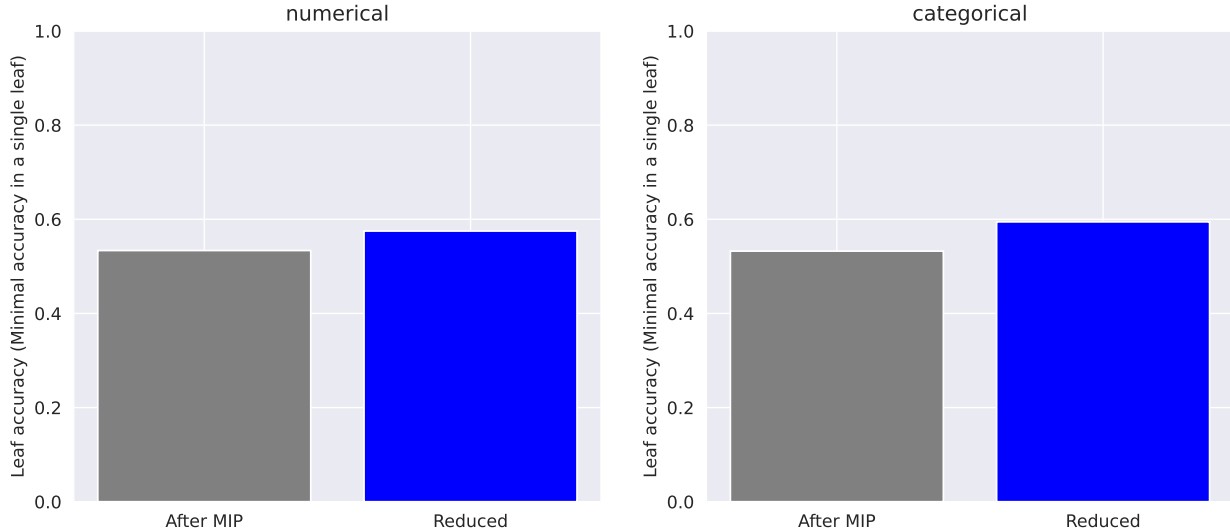

Figure 9: Effect of reduction on leaf accuracy of the model. In grey is the leaf accuracy before reduction, and in blue is the leaf accuracy after reduction. The plot shows mean accuracy over all datasets of a given type created by the proposed model.

In Figure 10, we further provide a comparison of the complexity of the created trees by comparing the distributions of the number of leaves (or potential explanations) provided by the method.

The maximum amount of leaves of a tree with depth 4 is 16. CART model has, on average, around 8 leaves after reduction. The proposed model's distribution is close to the distribution of CART models. When solving the MIO formulation directly, the distribution is severely shifted toward very small trees. Our proposed method uses a default CART solution to warmstart the search, which might explain the shape of the distribution compared to the direct method and CART.

### A.6 Hyperparameter search distributions

We needed to optimize hyperparameters for extending models and CART trees used for comparisons. We used Bayesian hyperparameter search for that purpose.

#### A.6.1 Extending XGBoost models

For the hyperparameter search of XGBoost models in leaves, we used the distributions listed in Table 6. The parameters are almost all the same as those used by (Grinsztajn et al., 2022). Only the Number of estimators and Max depth were more constrained to account for the fewer samples available for training.

The Bayesian optimization was run for 50 iterations, with 3-fold cross-validation in every leaf that contained enough points to perform the optimization. The same process was used to extend all tested trees.

In leaves with an insufficient amount of samples to perform the cross-validation (less than 3 samples of at least one class in our case), we train an XGBoost model with a single tree of max depth 5. In leaves with 100% training accuracy, we do not learn any model and use the majority class.

#### A.6.2 CART models

For the hyperparameter optimization of CART models, we also used Bayesian search, with the distributions shown in Table 7.

The search was run for 100 iterations, with 5-fold cross-validation on the same training data sets as our model. After this search, the best hyperparameters were used to train the model on the full training data.

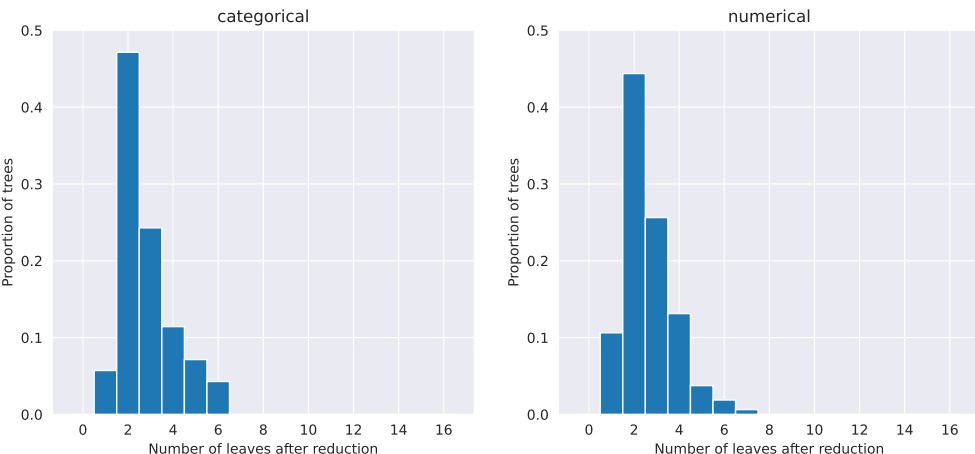

(a) Histogram of the number of leaves of the reduced trees optimized directly using the proposed formulation. The trees are heavily pruned.

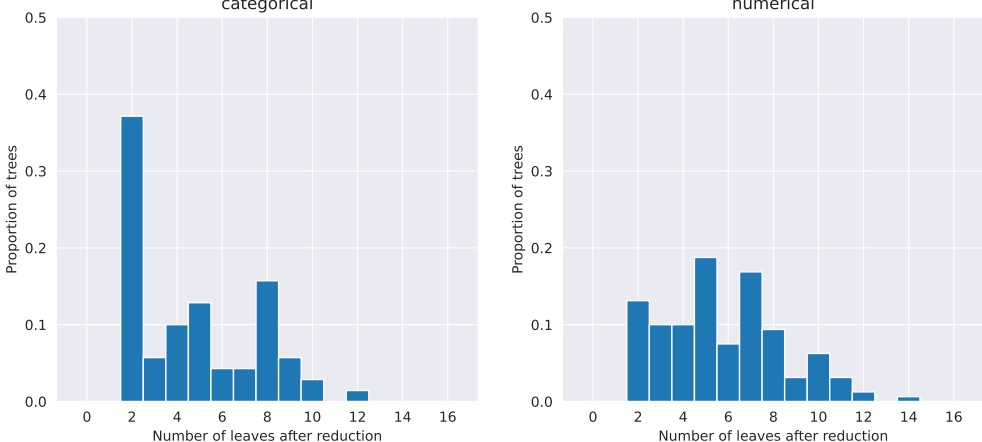

(b) Histogram of the number of leaves of the reduced trees created with the proposed formulation, warmstarted using a simple CART solution. The trees are smaller compared to well-optimized CART but retain some complexity. This was the chosen method.

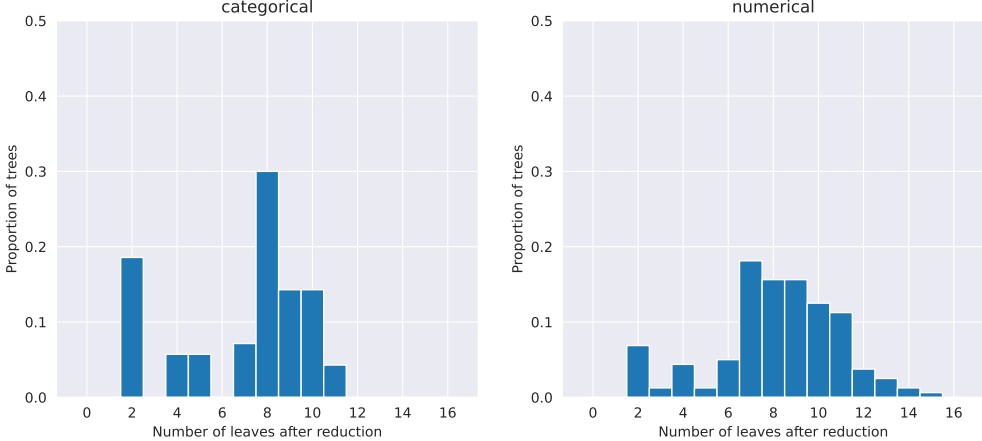

(c) Histogram of the number of leaves of the reduced trees created by CART with optimized hyperparameters.

Figure 10: Comparison of the numbers of leaves of trees after the reduction procedure.

Table 6: Distributions of hyperparameters of extending XGBoost models in leaves. These were used in the Bayesian hyperparameter search in each leaf separately. All distributions except Max depth and Number of estimators are the same as in (Grinsztajn et al., 2022). The two different distributions were selected smaller to improve the optimization time and to account for lower amounts of data.

| Parameter name | Distribution [range (inclusive)] |
|---|---|
| Max depth | UniformInteger [1, 7] |
| Number of estimators | UniformInteger [10, 500] |
| Min child weight | LogUniformInteger [1, 1e2] |
| Learning rate | Uniform [1e-5, 0.7] |
| Subsample | Uniform [0.5, 1] |
| Col sample by level | Uniform [0.5, 1] |
| Col sample by tree | Uniform [0.5, 1] |
| Gamma | LogUniform [1e-8, 7] |
| Alpha | LogUniform [1e-8, 1e2] |
| Lambda | LogUniform [1, 4] |

Table 7: Distributions of hyperparameters of CART models used to compare to our method. Max depth and Min samples in a leaf were fixed, but remain in the table for completeness.

| Parameter name | Distribution [range (inclusive)] |
|---|---|
| Max depth | UniformInteger [4, 4] |
| Min samples split | UniformInteger [2, 100] |
| Min samples leaf | UniformInteger [50, 50] |
| Max leaf nodes | UniformInteger [2, 16] |
| Min impurity decrease | Uniform [0, 0.2] |
| Cost complexity pruning parameter $\alpha$ | Uniform [0, 0.3] |

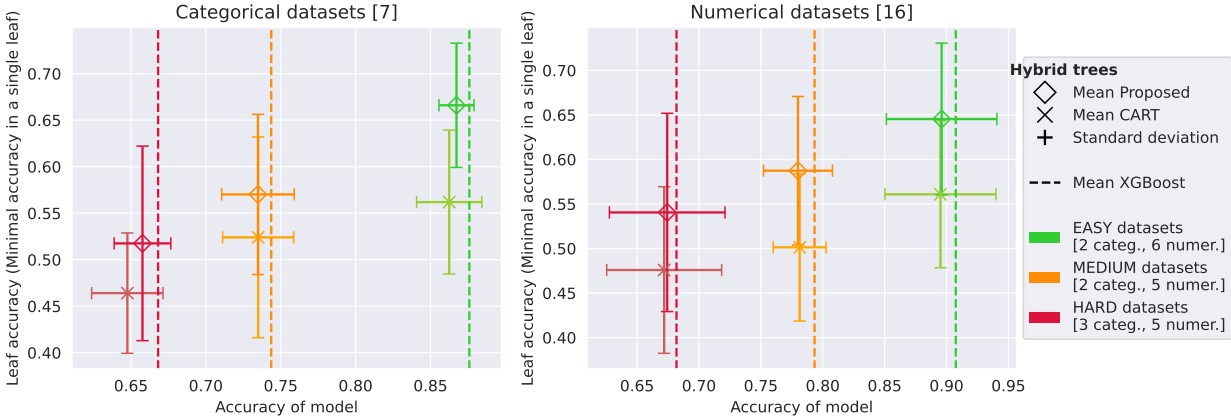

Figure 11: Results on out-of-sample data on all classification datasets from the tabular benchmark partitioned into 3 categories by complexity. In square brackets are the numbers of datasets belonging to each partition. This plot shows that our method, when extended in leaves, does not significantly decrease overall performance compared to pure XGBoost while sometimes improving upon accuracy obtainable by extended CART. And it does so consistently for datasets of varying complexity.

Table 8: Categorical datasets. Mean accuracy of models on out-of-sample data and average ranks.

|  | Leaf Accuracy | | Hybrid-tree Accuracy | | |
| --- | --- | --- | --- | --- | --- |
| **categorical datasets** | CART | Proposed | CART | Proposed | XGBoost |
| albert | 0.5033 | **0.5706** | 0.6455 | 0.6510 | **0.6559** |
| compas-two-years | 0.4504 | **0.5711** | 0.6714 | 0.6772 | **0.6807** |
| covertype | 0.5966 | **0.7071** | 0.8482 | 0.8567 | **0.8658** |
| default-of-credit-card-clients | 0.4471 | **0.5246** | 0.7110 | 0.7117 | **0.7184** |
| electricity | **0.6392** | 0.6250 | 0.8859 | 0.8781 | **0.8861** |
| eye_movements | **0.4202** | 0.4109 | 0.6303 | 0.6449 | **0.6677** |
| road-safety | 0.5701 | **0.6158** | 0.7573 | 0.7579 | **0.7689** |
| Mean rank | 1.7143 | **1.2857** | 2.8571 | 2.1429 | **1.0000** |

The resulting tree was reduced, and every leaf was extended by an XGBoost model in the same way as our models.

## A.7 Detailed results

Figure 11 shows again the average performance separately on categorical and numerical datasets divided into three groups by complexity. The complexity measure is based on the performance of XGBoost provided by the benchmark authors. The thresholds of partitions are 0.7 and 0.8 for datasets containing categorical features and 0.75 and 0.85 for datasets with only numerical features. The thresholds were selected in order to separate too easy and too hard datasets, which make the plots less informative, and to explore behavior on datasets with varying inner complexity. We see that the proposed method always significantly improves the leaf accuracy compared to CART.

We also provide the full results for each dataset. Figures 12 and 13 are decomposed variants of Figure 3 for categorical and numerical datasets, respectively. We also provide exact results in Tables 8 and 9, respectively. The detailed results show that the proposed model outperforms the CART model in both accuracy measures on almost all datasets and has comparable accuracy to XGBoost. Performed statistical tests (signed test and Wilcoxon's signed-rank test) resulted in proving the statistical significance of the better performance of the proposed model, compared to CART.

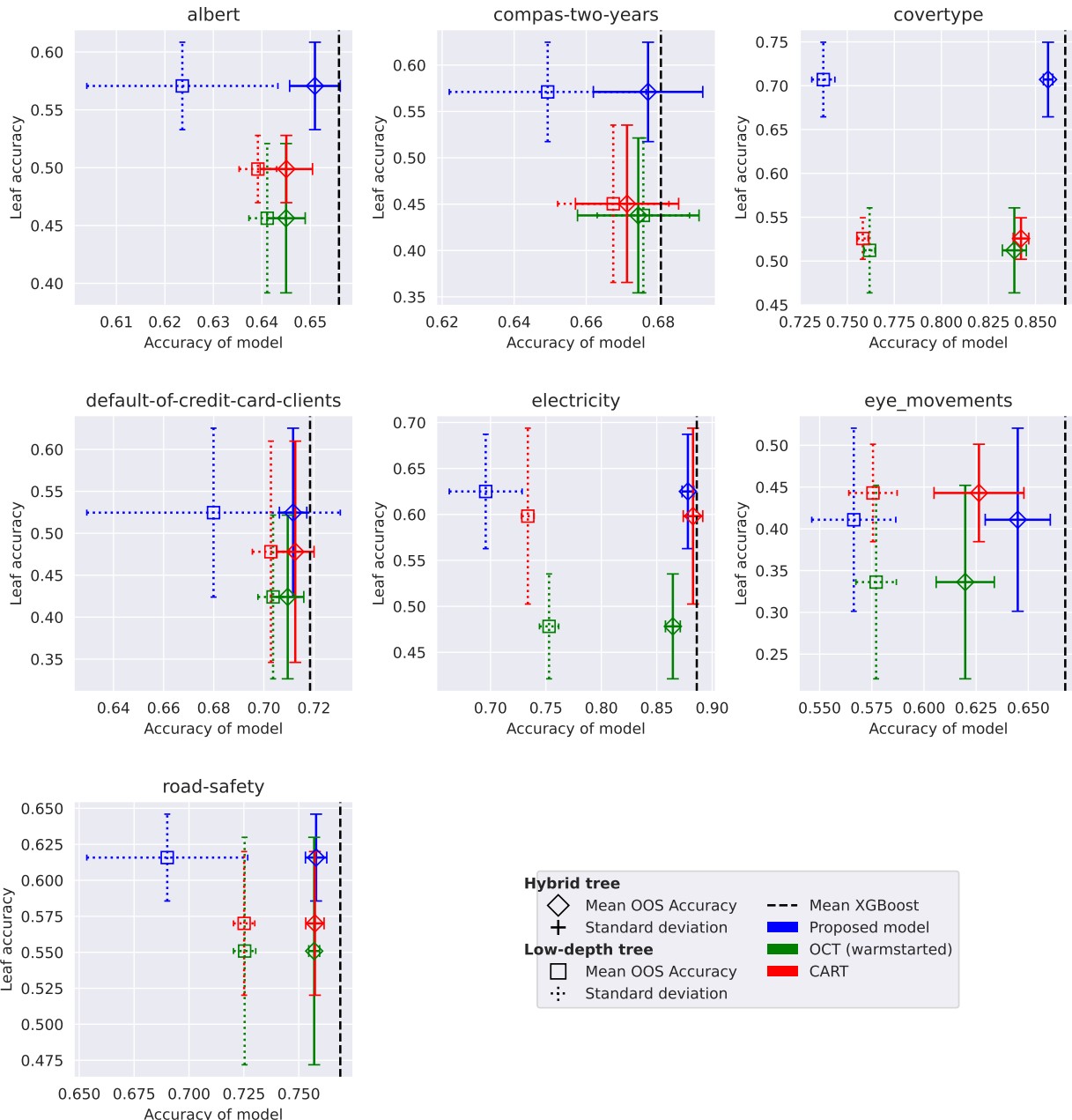

Figure 12: Detailed performance comparison of our model on categorical datasets.

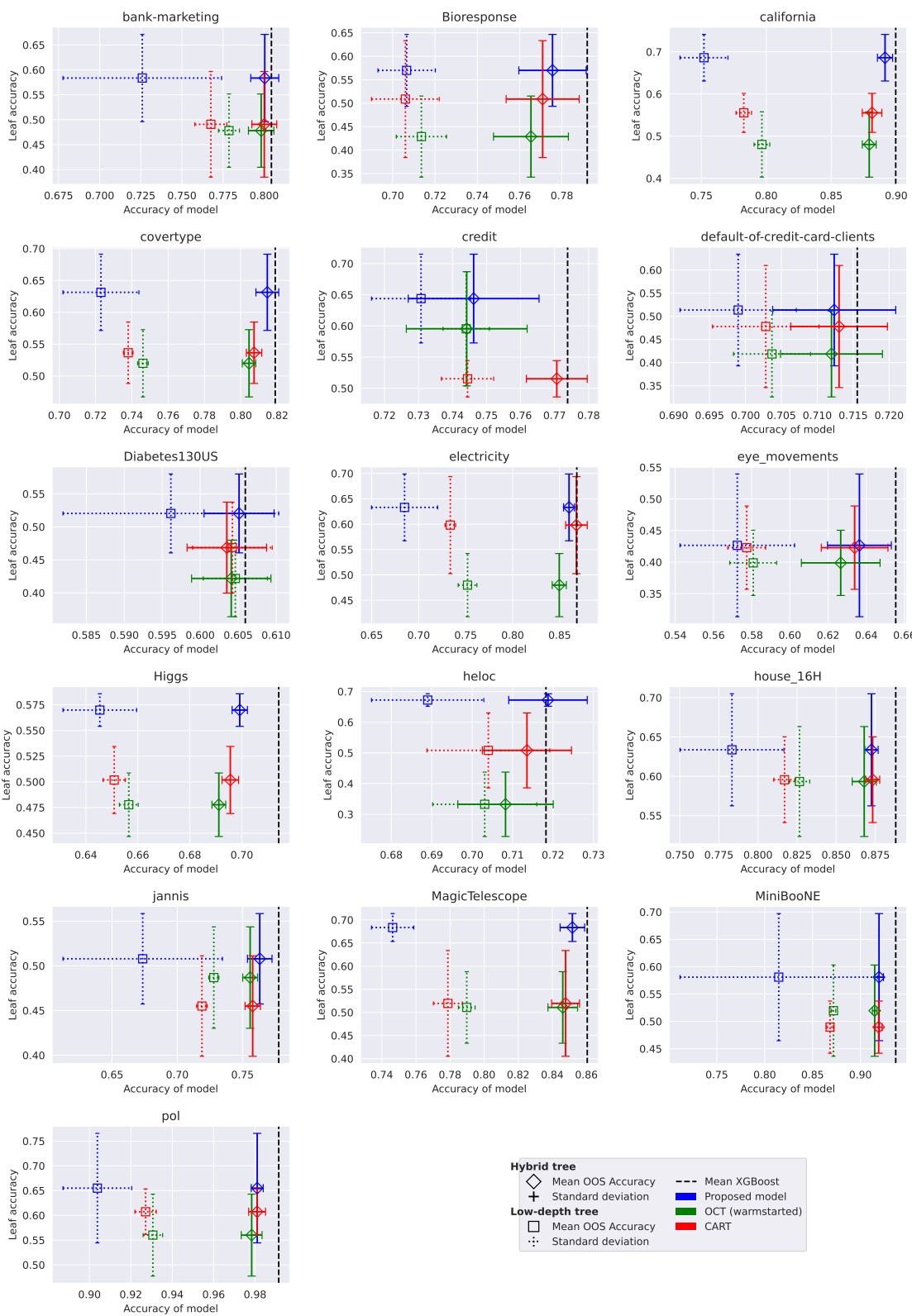

Figure 13: Detailed performance comparison of our model on numerical datasets.

Table 9: Numerical datasets. Mean accuracy of models on out-of-sample data and average ranks.

| | Leaf Accuracy | | Hybrid-tree Accuracy | | |
| numerical datasets | CART | Proposed | CART | Proposed | XGBoost |
|---|---|---|---|---|---|
| bank-marketing | 0.4861 | **0.5837** | 0.8001 | 0.8003 | **0.8044** |
| Bioresponse | 0.5201 | **0.5700** | 0.7702 | 0.7755 | **0.7920** |
| california | 0.5593 | **0.6861** | 0.8827 | 0.8914 | **0.8997** |
| covertype | 0.5365 | **0.6314** | 0.8074 | 0.8147 | **0.8190** |
| credit | 0.5153 | **0.6439** | 0.7707 | 0.7462 | **0.7738** |
| default-of-credit-card-clients | 0.5011 | **0.5136** | 0.7132 | 0.7124 | **0.7156** |
| Diabetes130US | 0.4630 | **0.5204** | 0.6028 | 0.6051 | **0.6059** |
| electricity | **0.6392** | 0.6331 | **0.8724** | 0.8600 | 0.8683 |
| eye_movements | 0.4229 | **0.4265** | 0.6343 | 0.6364 | **0.6554** |
| Higgs | 0.4910 | **0.5698** | 0.6953 | 0.6992 | **0.7142** |
| heloc | 0.4881 | **0.6722** | 0.7128 | **0.7188** | 0.7183 |
| house_16H | 0.5956 | **0.6336** | 0.8733 | 0.8726 | **0.8881** |
| jannis | 0.4550 | **0.5079** | 0.7579 | 0.7632 | **0.7778** |
| MagicTelescope | 0.5168 | **0.6835** | 0.8478 | 0.8518 | **0.8605** |
| MiniBooNE | 0.4821 | **0.5809** | 0.9192 | 0.9194 | **0.9369** |
| pol | 0.6073 | **0.6550** | 0.9810 | 0.9811 | **0.9915** |
| Mean rank | 1.9375 | **1.0625** | 2.6875 | 2.1875 | **1.1250** |

## A.8 Fairness aspect

When constructing a decision tree using demographic data, the leaves will correspond to (possibly protected) subgroups of people. An explanation provided to a subgroup classified by a leaf with low accuracy is less useful, compared to the explanations other subgroups obtain.

Measuring explanation fairness via leaf accuracy falls into the category of Value-based Explanation Fairness ($\Delta_{\mathbf{VEF}}$) of Zhao et al. (2023). Maximizing the leaf accuracy then minimizes the absolute difference of accuracy between leaves (i.e., explanations), corresponding to the definition of $\Delta_{\mathbf{VEF}}$ (Zhao et al., 2023).

Therefore, in addition to the benchmark (Grinsztajn et al., 2022), we evaluate the method on folktables (Ding et al., 2021), a fairness benchmark. The data contains demographic information from the American Community Survey for the year 2018. In our case we take data for 5 US states (California, New Mexico, Florida, New York, and Louisiana) and test using the 5 tasks pre-defined in the library (ACSIncome, ACSTravelTime, ACSPublicCoverage, ACSMobility, ACSEmployment) (Ding et al., 2021).

See Figure 14 for aggregate results (or Figure 15 for details). Here, the OCT still underperforms in leaf accuracy and hybrid-tree accuracy, but CART has slightly better model accuracy when extended with the XGBoost models. Nonetheless, the proposed method outperforms both other in terms of leaf accuracy by a significant margin.

## A.9 Other optimization approaches

The best-performing approach of warmstarting the MIO solver with a CART solution is not the only one we tested. In Figure 16, we see a comparison of three different approaches to optimization.

- *Direct* refers to the straightforward use of the MIO formulation.

- *Warmstarted* uses a simple CART solution (created using default hyperparameters) as a starting point of the solving process.

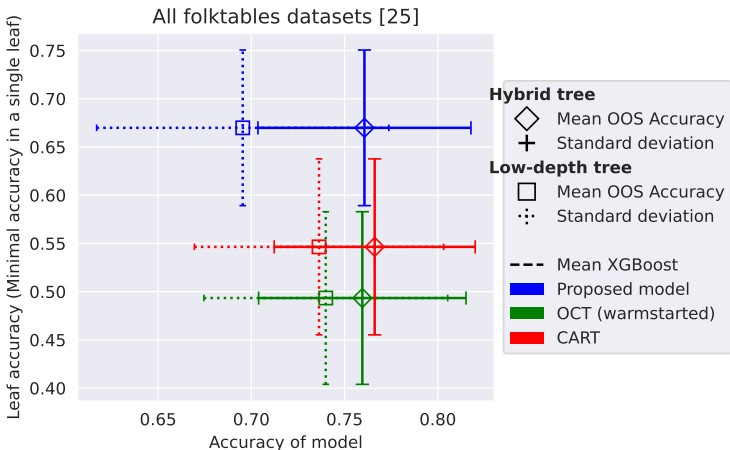

Figure 14: Aggregate comparison of tested approaches on data from folktables (Ding et al., 2021).

Table 10: Comparison of Gradual and Warmstarted approach. Positive numbers show an advantage in the mean accuracy of the Proposed (Warmstarted) approach. Gradual refers to the approach when the depth of the tree is gradually increased during the optimization process.

|  | Data type | Minimal | Mean ($\pm$ std) | Maximal |
|---|---|---|---|---|
| **Leaf Accuracy** | categorical | $-0.1094$ | $0.0122 \pm 0.0753$ | $0.1130$ |
|  | numerical | $-0.0867$ | $0.0117 \pm 0.0624$ | $0.1154$ |
| **Hybrid-tree Accuracy** | categorical | $-0.0219$ | $-0.0021 \pm 0.0094$ | $0.0083$ |
|  | numerical | $-0.0103$ | $-0.0023 \pm 0.0056$ | $0.0076$ |

- *Gradual* refers to a special process where we start by training a tree with a depth equal to 1 and use the solution found in some given time to start the search for a tree with a depth of 2, and so forth until we reach the desired depth.

Interestingly, while the direct approach understandably does not reach a performance similar to the warm-started variant, the gradual approach shows more promise. It has higher hybrid-tree accuracy by another 0.2 percentage points on average while having lower leaf accuracy by about 1.2 percentage points compared to the warmstarted approach (cf. Table 10).

### A.10   Warmstarting models

The proposed model uses simple CART solutions to warmstart the MIO solver. This is a common procedure that can provide an initial bound on the quality of a solution and thus improve the performance of the branch-and-cut algorithm that is central to MIO solving.

The final solutions obtained by the proposed method (within the given time budget) differ significantly from those used in the warm start. See Table 11 for an overview of mean accuracies of the models. The performance of the warm-start models is similar to that of the tuned CART methods.

In addition, see Section A.11.1 for an example comparison of a warm-start model with the final optimized model. While anecdotal, it showcases a case with a clearly distinct model in both structure and in the used thresholds. All nodes except for the root are distinct.

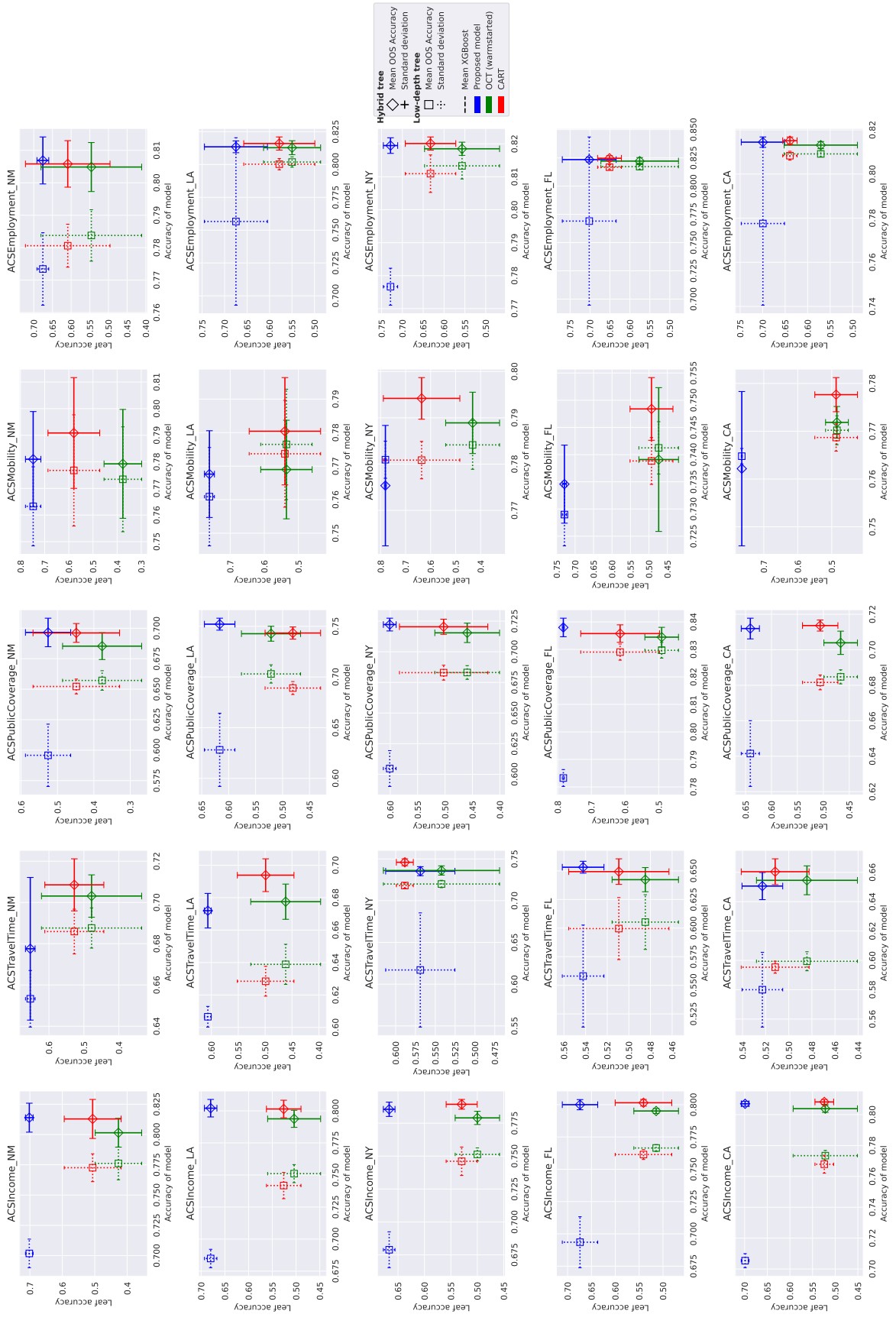

Figure 15: Detailed results on the folktables data for the 5 folktables tasks (columns) and 5 US States (rows).

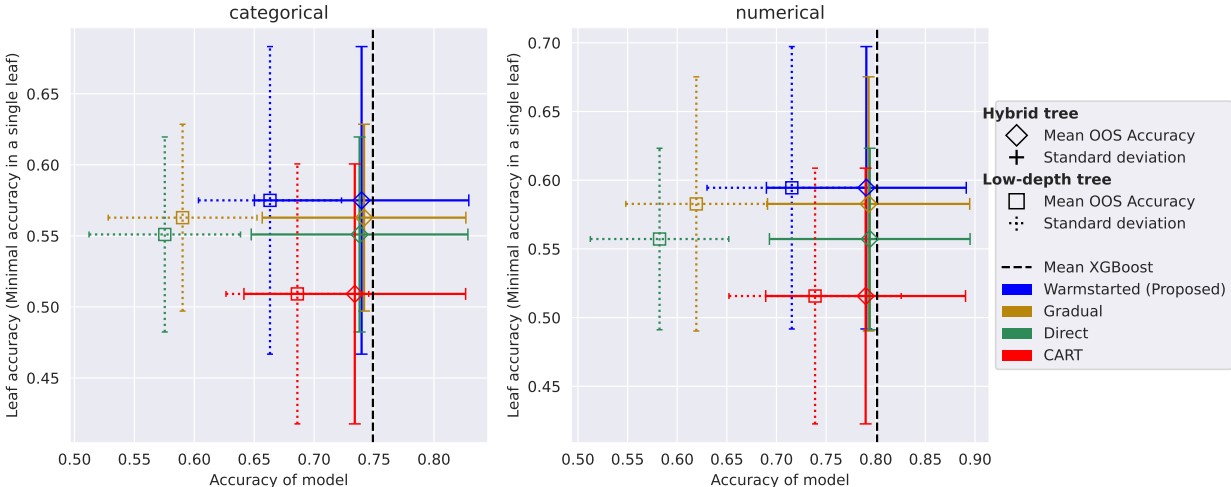

Figure 16: Comparison of various approaches to using the formulation, given the same resources and conditions. Direct means solving without warmstarting. Gradual solves the formulation with increasing depth, using the previous solution to find a deeper model.

Table 11: Mean accuracy over datasets (each dataset averaged over seeds first).

| Metric | Our model | CART used in warm start |
|---|---|---|
| Shallow tree accuracy | 69.95 | 72.33 |
| Leaf accuracy | 58.86 | 49.18 |
| Hybrid tree accuracy | 77.31 | 76.67 |

## A.11   Agreement with extending model

We now investigate the question of how much of the data is explained by the shallow trees. When using the hybrid tree structure as a partially interpretable model, we find that around 81% of the data retains its label after applying the XGBoost models. This means that the extensions change the overall decision in only around 19% of cases, and in some datasets even fewer than 10%, as shown in Figure 17.

This holds for extending the proposed models. In comparison, when extending CART trees used for warmstarting, they have around 5 percentage points higher agreement rates, namely 86.87%. This discrepancy is expected, given their higher model accuracy prior to extension.

### A.11.1   Case study

In addition to the aggregated results, we also investigate a single case study. We use a single split of the compas-two-years dataset, which represents the task of predicting whether an individual reoffends within 2 years. See Figure 18 for visualization.

In terms of model agreement, we compute agreement as the proportion of test samples that retain the class assigned by the leaf (and shown in the Figure) after being processed by the extending model. Notice that there is a non-trivial number of leaves with 100% agreement rate, meaning that the explanation holds for each sample from the test set in the given leaf. Overall, much like in the aggregated case, the CART solution (which, on this data split, is the same as the CART used for warmstarting) has a notably higher agreement rate, despite having more leaves with some disagreements. Those disagreements are limited in absolute numbers and appear larger only in relative terms. At the same time, the usefulness of an explanation of a leaf with 55% agreement rate might also be considered questionable.

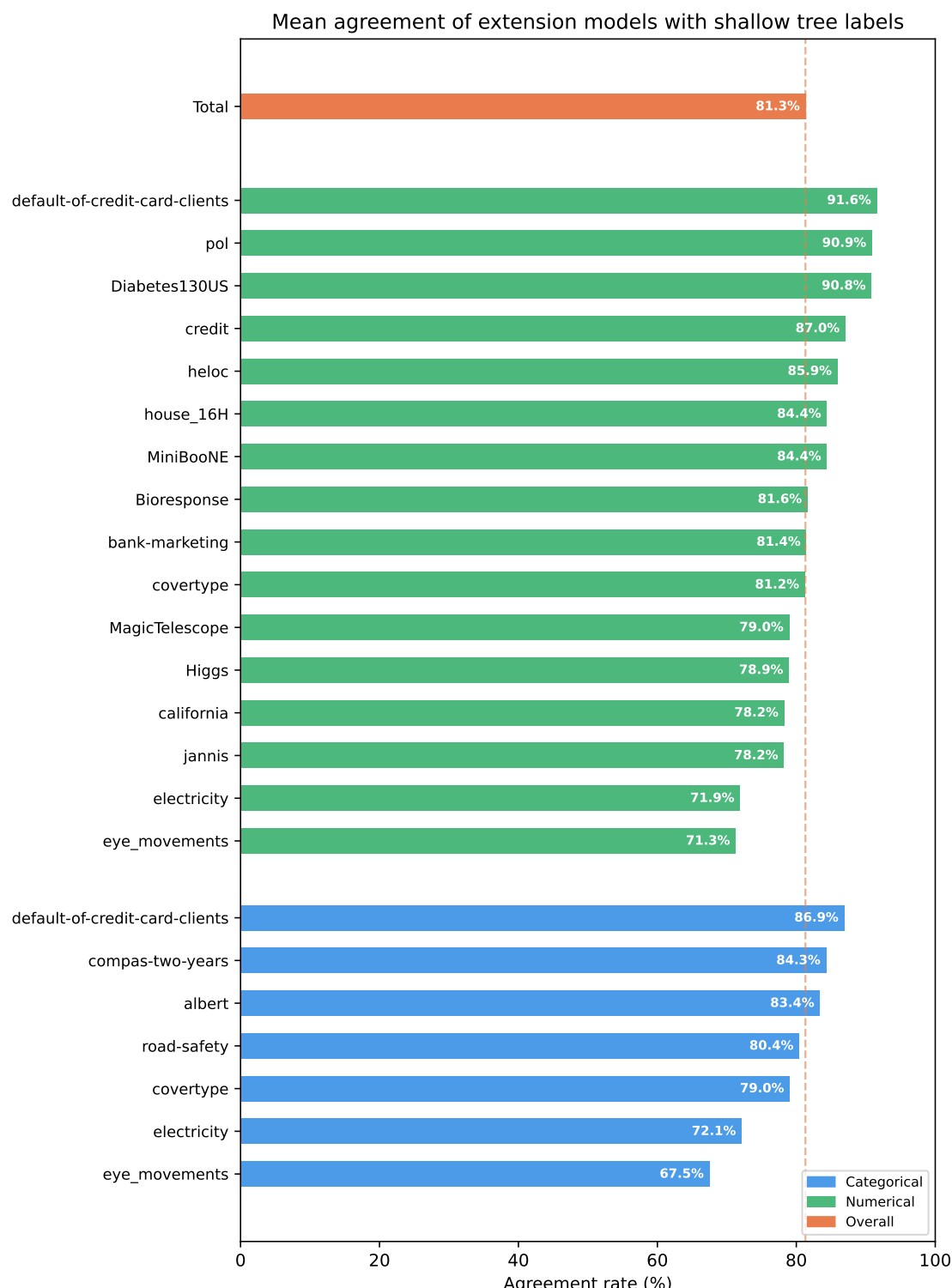

Figure 17: Per dataset comparison of extension model agreement rates. The proportion of labels on which the shallow tree agrees with the extending model.

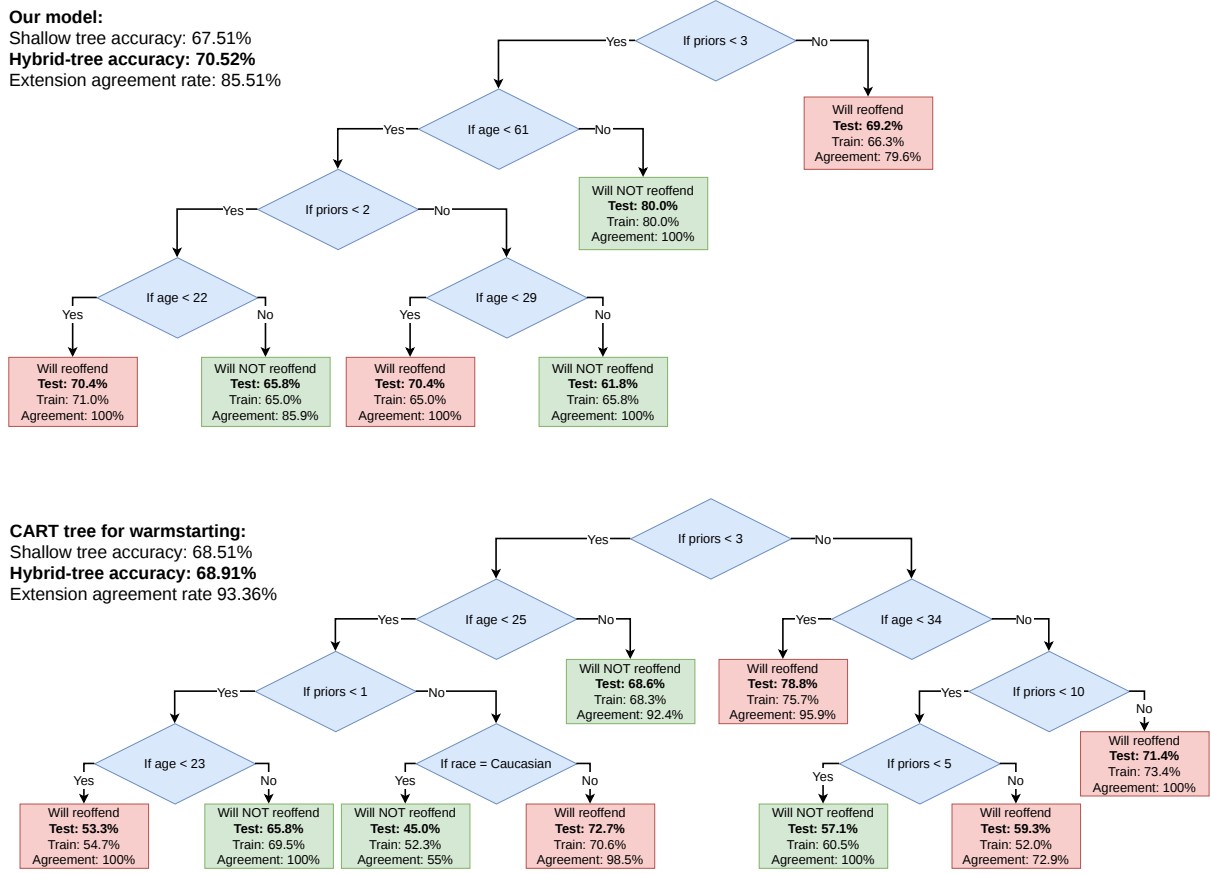

Figure 18: Comparison of trees created by our approach and the tree used for warmstarting the solver. We include the per-leaf proportion of samples that the extending model classifies as the same class. Train and test accuracies are computed before the extension.

From a fairness perspective, we notice that the least accurate leaf is created directly by a condition on race, which classifies all Caucasians as non-offending, with very poor accuracy (52.3% on training data) that did not generalize well (45% on test set). If this tree were used as a global explanation of a given black-box model, one might come to believe that the underlying model favors Caucasians among young individuals with 2 priors. Using leaf accuracy, we can see that this rule has little support in the data, and thus its validity is weak. Therefore, the usefulness of this explanation tree could be called into question.

## A.12 Ablation Analyses

We provide some comparison experiments performed by changing a single hyperparameter (or a few related ones) and comparing the performance.

### A.12.1 Unlimited depth CART

An argument could be made against our choice to compare our method to CART trees with the same limit on depth. Figure 19 and Table 12 in more detail show a comparison of CART models with a maximal depth of 4 and a maximal depth of 20. The actual depth limit for each model was optimized along with other hyperparameters using the Bayes hyperparameter optimization procedure.

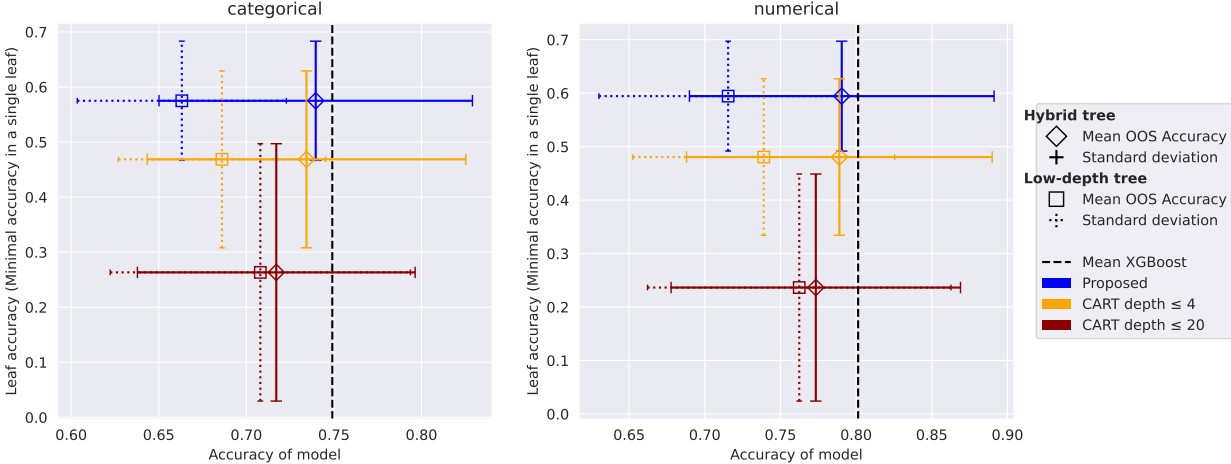

Figure 19: Comparison of CART tree results with limited depth and without such a strict limit on the depth. Deeper trees provide worse explanations (due to the length of explanation) and perform worse in both accuracy measures. For a more detailed description of the differences introduced by the depth, consult Table 12.

Table 12: Detailed view of the differences in the accuracy between CART trees with max depth 4 and CART trees with max depth 20. A positive number means the accuracy advantage of the more constrained model (depth $\leq 4$). For a graphical representation, see Figure 19.

|  | Data type | Minimal | Mean ($\pm$ std) | Maximal |
|---|---|---|---|---|
| **Leaf Accuracy** | categorical | $-0.0769$ | $0.2053 \pm 0.2389$ | $0.5404$ |
|  | numerical | $-0.1183$ | $0.2441 \pm 0.2115$ | $0.5680$ |
| **Hybrid-tree Accuracy** | categorical | $-0.0025$ | $0.0173 \pm 0.0185$ | $0.0420$ |
|  | numerical | $-0.0006$ | $0.0156 \pm 0.0119$ | $0.0370$ |

Note that these tests were performed in earlier stages of testing without a fixed lower bound on the number of samples in a leaf and without cost complexity pruning. The lower bound on the number of samples was optimized using the Bayes optimization in the range $[0, 50]$.

The aggregated results show worse performance regarding both leaf accuracy and hybrid-tree accuracy. Not only do the deeper trees perform worse, but the length of provided explanations is also well above the 5-9 threshold suggested as the limit of human understanding (Feldman, 2000).

### A.12.2 Different minimum number of samples in leaves

A similar comparison is to see the performance of classically optimized lower bound on the number of samples in each leaf. Figure 20 shows a comparison of CART models when the lower bound is fixed to 50 and when it is optimized within the range from 1 to 60 using Bayesian hyperparameter optimization. The figure also includes the performance of the proposed model when $N_{\min}$ is set to 1.

It stresses the importance of setting a minimal amount of samples in leaves. Without enough points to support the leaf's accuracy, it is more likely to be overfitted. On the other hand, when choosing the $N_{\min}$ parameter too high, we restrict some possibly beneficial splits, supported by a smaller amount of training data.

$N_{\min}$ is a critical hyperparameter, and further testing could provide more insight into the proposed model's performance.

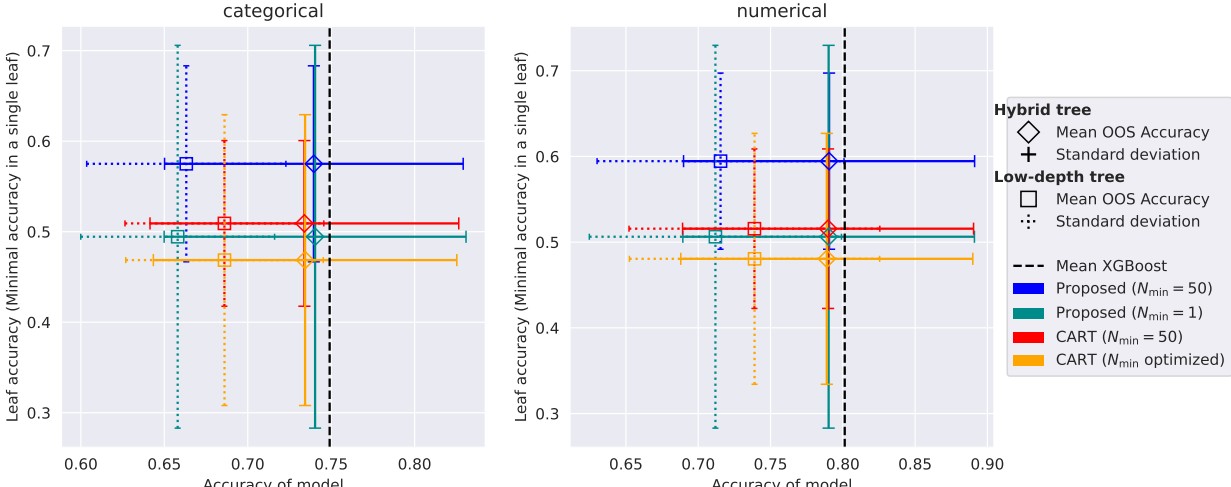

Figure 20: Comparison of performance of the proposed model with minimum samples in leaves equal 50 and 1 and of CART trees with parameter equivalent to $N_{\min}$ fixed to 50 or undergoing hyperparameter optimization. Low values of $N_{\min}$ lead to overfitting to training data and worse out-of-sample performance. Notice the high deviation of the model with $N_{\min} = 1$. CART trees suffer from a similar thing, which suggests two things. Hyperparameter optimization does not opt for high $N_{\min}$ values, and this seems to be a property of trees in general, no matter how they are obtained.

### A.12.3   Non-warmstarted OCT

We compare our method to warmstarted OCT because the proposed method also starts from the same initial CART solution. This makes them more comparable. However, we also tested the OCT variant directly optimized from the MIO formulation. See the results in Figure 21. Both OCT models were run with the same hyperparameters as the proposed model. Those being the heuristics-oriented solver, depth equal to 4, and a minimal amount of samples in leaves equal to 50.

The average OCT performs worse than all our approaches (cf. Figure 16, all approaches are above the 0.55 mark, contrary to OCT in Figure 21), but the improvement from the warmstarted variant is intriguing since it clearly manages to overtake the CART model. Especially considering that it is not caused by the direct OCT method's inability to create complex trees without warmstarting. This is supported by Figure 22 showing a distribution of the number of leaves similar to the distribution of CART trees (cf. Figure 10). This suggests that the OCT trees have comparable tree complexity to CART and provide more useful explanations than CART, even without our extension to the formulation. This is an interesting result, considering the fact that neither CART nor OCT methods optimize for leaf accuracy.

Our model, however, more than doubles the improvement of direct OCT.

### A.12.4   Deeper trees

Lastly, we provide a comparison of the proposed model of depths 4 and 5. Figure 23a shows better overall results for shallower trees. This is likely caused by the exponential increase in memory requirements, given the decrease in overall accuracy as well. We provide data about its memory usage in Figure 23. With a model of twice the complexity, the solver struggles to achieve comparable results to the shallower proposed model.

This is certainly a topic of further exploration by incorporating scalability improvements proposed in the literature.

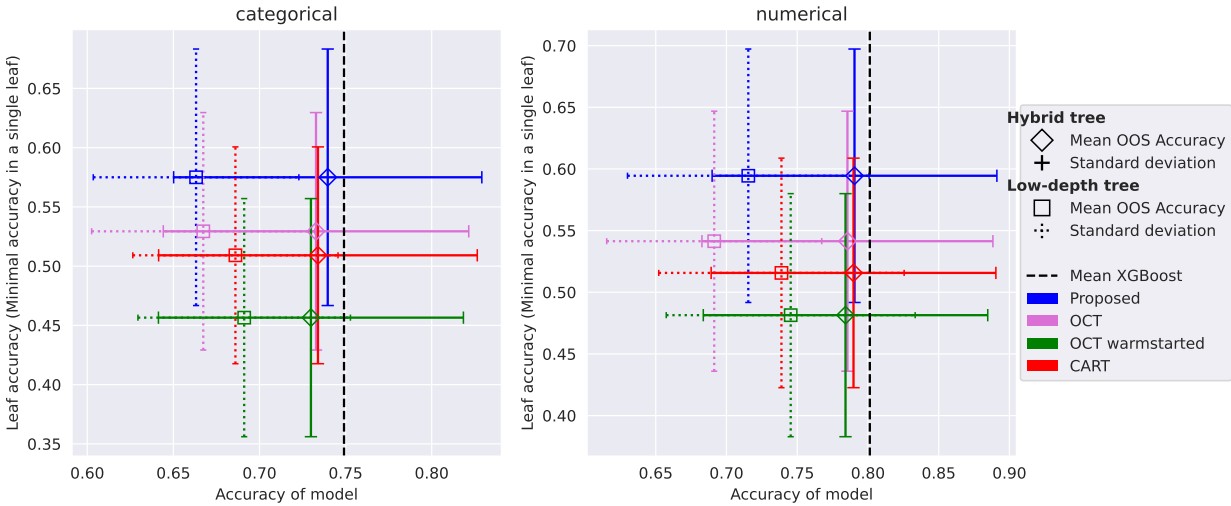

Figure 21: Comparison of OCT trees that are warmstarted the same way as our Proposed model and OCT without the warmstart, optimized directly. Interestingly, direct OCT performs significantly better.

### A.13 More data

The 10,000 size limit on training samples was suggested by the authors of the benchmark (Grinsztajn et al., 2022). Another good reason for such a limit is that we want our model to balance the size of the formulation and the capability of the formulated model. In other words, if we take a small amount of data, we are less likely to grasp the intricacies of the target variable distribution within the dataset. And if we take too many samples, we create a formulation that will not achieve good performance in a reasonable time.

In a comparison of a model learned on a training dataset limited to 10,000 samples with a dataset limited to 50,000 samples, we saw that more data does not necessarily lead to a better model, given the same time resources. The 50,000 sample model performed worse in terms of leaf accuracy due to the overly complex formulation. On the other hand, it improved the overall model accuracy, which was unsurprising since each leaf contained more samples.

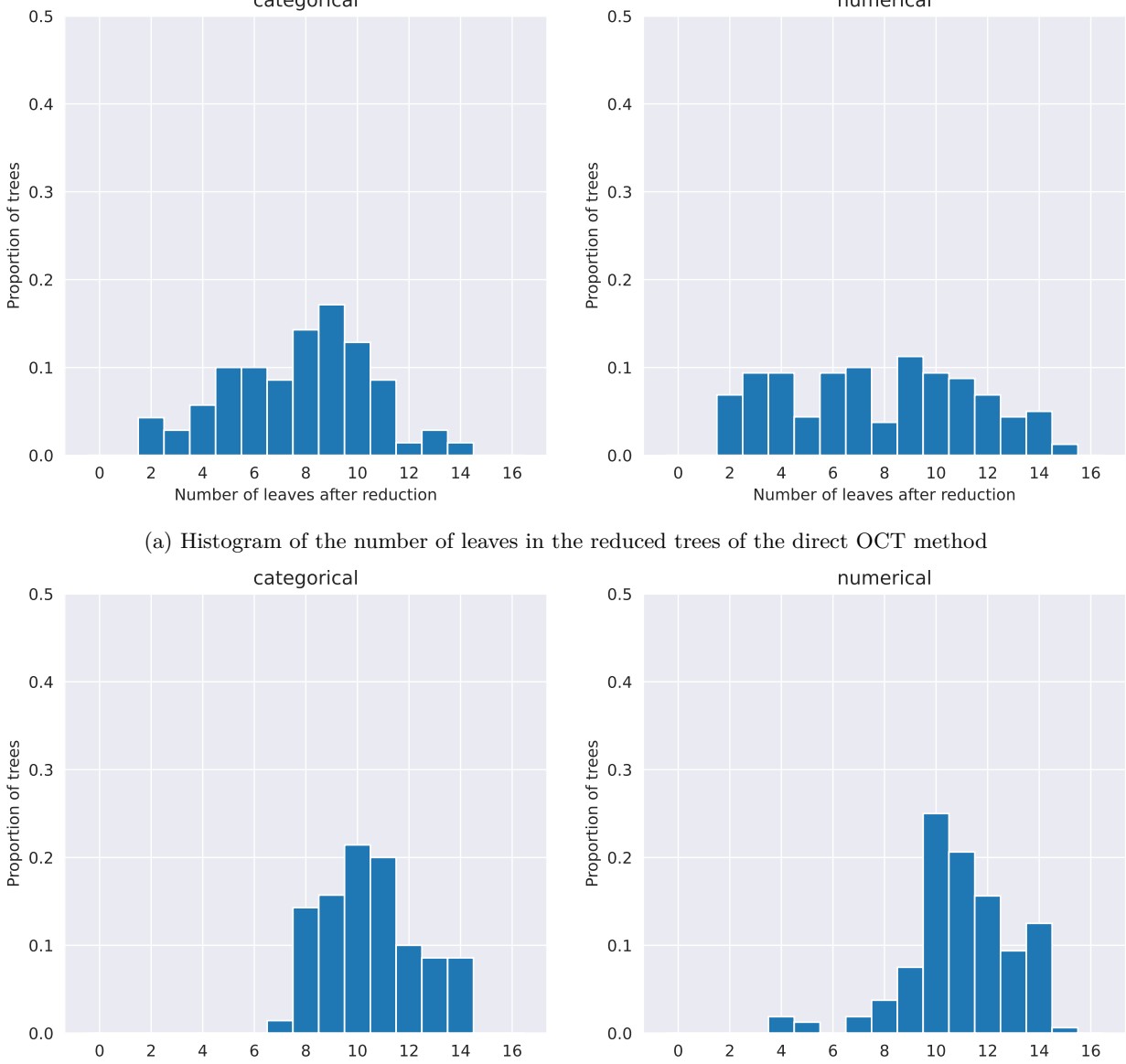

(a) Histogram of the number of leaves in the reduced trees of the direct OCT method

(b) Histogram of the number of leaves in the reduced trees of the warmstarted OCT method.

Figure 22: Comparison of reduced tree complexity of the OCT with and without warmstart. OCT without warmstart creates trees of similar distribution as the CART method (cf. Figure 10). And it achieves better leaf accuracy than CART (cf. Figure 21) despite neither optimizing that objective.

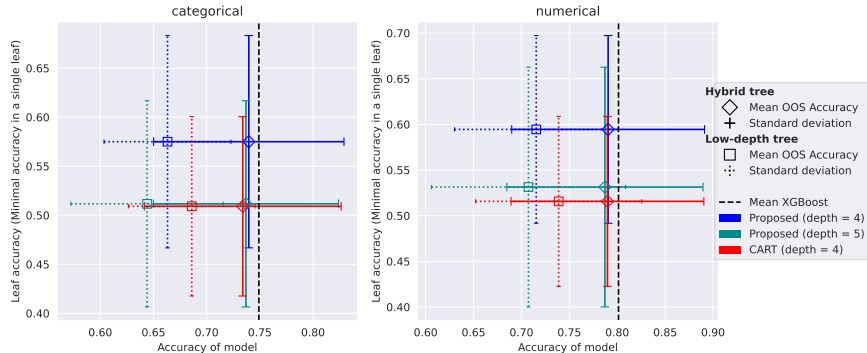

(a) Comparison of performances of the proposed model with depths 4 and 5. Shallower trees perform better, possibly because they are easier to optimize.

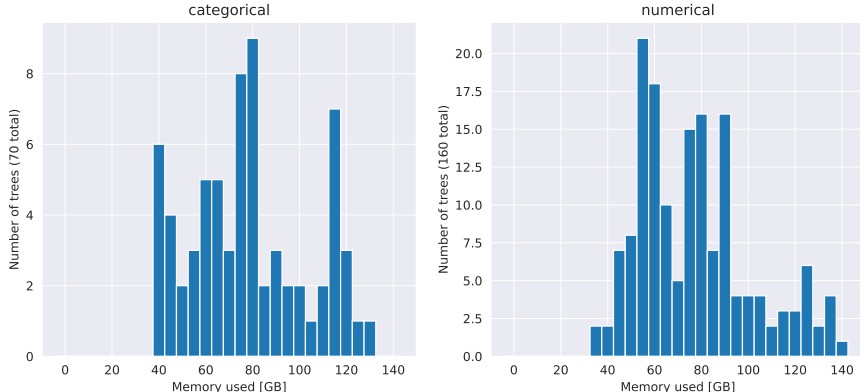

(b) Depth 5. Histogram of memory requirements of MIO solver for all dataset splits.

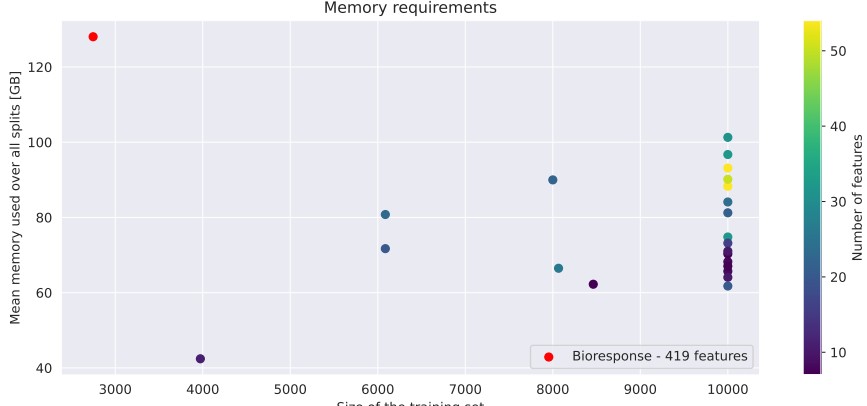

(c) Depth 5. Mean memory requirements on datasets. Dots are colored according to the number of features. Dataset Bioresponse is excluded from the color mapping due to having a significantly higher number of features. Training sets were clipped to a maximum of 10,000 points.

Figure 23: Comparison of the memory requirements of the Proposed model with depth 5. The mean memory requirement almost increases from cca 51.1 GB to 77.3 GB with an increase in depth from 4 to 5. Compare the above plots with Figure 7.

