# OpenReview forum: "Improving the Usefulness of Decision Trees as Explanations"
_TMLR — Decision pending for TMLR_

### Review · Reviewer_qPHC · 2026-05-13

**Summary Of Contributions:**

This paper studies an important problem of improving the usefulness of decision trees, providing more accurate explanations. Generally, this paper is well-written and easy to follow. The method is technically sound. Experiments show the effectiveness of the proposals to some extent.

**Audience:**

No

**Audience Explanation:**

Although the decision tree is a classic topic in machine learning, it seems old-fashioned and may not attract a large audience.

**Claims And Evidence:**

No

**Claims Explanation:**

It would be better to include more theoretical analysis to demonstrate that the proposed method can improve the usefulness of decision trees in the main content. Further, more intuitive case studies are suggested to be included to show the effectiveness of the proposals. In addition, it would be better to clarify the importance of this paper in the current machine learning community.

**Requested Changes:**

Please see the above comments.

---

> ### Author Response · Authors · 2026-06-05
> **Rebuttal**
>
> We thank the reviewer for their assessment. We are pleased that they find the topic important, the paper well written, the method sound, and the experiments consistent with the proposal.
>
> The reviewer raises three concerns, which we address below:
>
> *Theoretical analysis* - We note that it is unclear to us what form of theoretical analysis the reviewer has in mind. We do not provide any theoretical results; the paper argues for optimizing the accuracy of the least accurate leaf, defined by Eq. 1. We motivate leaf accuracy as a measure of the usefulness of the tree as an explanation in the Introduction and showcase that it can be improved using the proposed algorithm by empirical comparison. We have strengthened the argument linking leaf accuracy and usefulness in the Introduction and added a case study showcasing the relationship in A.11.1.
>
> *Case studies* - We provide a case study in Figure 1, showing that our method can generate trees without low-accuracy leaves while maintaining comparable predictive performance. In addition, we have added another case study in Appendix A.11.1. There, CART creates a node separating Caucasians and other races, while the leaf has 45% accuracy. If the tree were a global explanation of a black-box model, the leaf would thus suggest that Caucasians are favored, even though the data does not support that. A method creating such misleading explanations is thus arguably less useful.
>
> *Current ML relevance* - We would respectfully disagree with the assessment that no individuals in TMLR’s audience would be interested in the paper’s findings. Decision trees are the workhorse of data science (e.g., in financial-services applications, they are required by the regulators; in other cases, they are used because of their speed). Correspondingly, TMLR publishes research on decision trees; recent examples include DT learning [1] or the use of Mixed-Integer Optimization for tree models [2].
>
> [1] Learning a Decision Tree Algorithm with Transformers,Yufan Zhuang, Liyuan Liu, Chandan Singh, Jingbo Shang, Jianfeng Gao, August 2024
>
> [2] Optimal Pattern Detection Tree for Symbolic Rule-Based Classification, Young-Chae Hong, Yangho Chen, April 2026

---

### Review · Reviewer_44nq · 2026-05-20

**Summary Of Contributions:**

This paper addresses a crucial but often overlooked aspect of global model interpretability: the uneven distribution of local accuracy within decision tree leaves. The authors argue that a global explanation (a shallow tree) is only as reliable as its weakest rule. To mitigate this, they propose a Mixed-Integer Optimization framework to train shallow decision trees by explicitly maximizing the minimum leaf accuracy. To bridge the inevitable gap in overall predictive performance caused by this constraint, the authors introduce a hybrid-tree approach where the leaves of this optimized shallow tree are extended with localized, black-box XGBoost models. They benchmark their approach against classic CART and Optimal Classification Trees across 23 tabular datasets and a fairness suite, demonstrating substantial gains in minimal leaf accuracy with a negligible sacrifice in overall accuracy compared to pure XGBoost.

Key Strengths:

1. Shifting the focus from global optimization criteria to a maximin leaf accuracy formulation directly addresses the reliability of local explanations in high-stakes domains.

2. Leveraging mixed-integer programming ensures an exact and globally optimized search space for the shallow tree, bypassing the pitfalls of greedy top-down splitting.

3. The empirical evaluation utilizes a well-established tabular benchmark from Grinsztajn et al. (2022) and covers both numerical and categorical data setups.

4. Linking the minimum leaf accuracy metric to Value-based Explanation Fairness highlights a strong ethical and regulatory utility.

Key Weaknesses:

1. The MIO formulation is computationally heavy, demanding up to 64 core-hours per split and failing to close the MIP gap.

2. Extending an explainable tree's leaves with complex black-box XGBoost models compromises the core claim of global interpretability, particularly given the high disagreement rates.

3.  The baseline comparisons suffer from inconsistent tuning or structural assumptions that subtly favor the proposed method.

**Audience:**

Yes

**Audience Explanation:**

Optimizing tree-based architectures for uniform sub-group fidelity is highly relevant to researchers working in Explainable AI, Fairness in Machine Learning, and Operations Research. The definition of leaf accuracy as a criterion for global explanation fairness aligns directly with current regulatory pressures that require high-stakes tabular models to be interpretable and non-discriminatory across demographics. The mathematical strategy of augmenting mathematical programming with Post-Hoc gradient boosting will spark meaningful discussions about how to balance sample efficiency, computational constraints, and rigorous performance thresholds.

**Broader Impact Concerns:**

The paper includes a helpful discussion on fairness benchmarks and links leaf accuracy to Value-based Explanation Fairness. However, there is a minor ethical risk that requires attention. Because the hybrid model embeds black-box models inside a shallow, seemingly "safe" decision tree, it runs the risk of providing an illusion of transparency to auditors or regulatory bodies. If a financial institution uses this method to justify credit scoring, it might present the shallow tree to regulators while the hidden XGBoost models drive the actual rejections. The authors should briefly add a sentence in their conclusion or impact statement warning against using the shallow tree as an audit trail when disagreement rates with the underlying leaf models are high.

**Claims And Evidence:**

No

**Claims Explanation:**

While the empirical results show clear improvements in the specific metric defined, several core claims regarding interpretability and methodological consistency are undercut by the design choices and empirical disclosures:

1. The authors claim that their model provides a more useful global explanation. However, in Section 5, they admit that the localized XGBoost models disagree with the shallow tree's predictions in 37.7% of cases on average. If a black-box model overwrites the logical rule of a leaf more than one-third of the time, the shallow tree ceases to be an accurate global explanation of the operational model. The claim that it remains useful as an explanation is misleading because a user cannot trust whether the simple rule or the hidden black-box dynamics determined their specific outcome.

2. The paper heavily relies on the global optimization capabilities of MIO to avoid greedy local minima. Yet, the Gurobi solver leaves an enormous 60% average MIP gap after 8 hours of heavy computation. This implies that the model being evaluated is not actually the mathematically optimal tree intended by the formulation, but rather a heuristically found feasible point highly influenced by the CART warmstart.

3. The comparison to OCT (Bertsimas & Dunn, 2017) shows that direct (non-warmstarted) OCT achieves significantly better leaf accuracy than warmstarted OCT, yet the authors choose to focus on the warmstarted variant in their main plots to assert dominance. For the 50,000-sample ablation study, the proposed model’s performance degrades due to optimization limits, but the authors contrast it against an XGBoost baseline trained on only 10,000 samples, making the comparative claim invalid.

**Requested Changes:**

1. The authors must explicitly reframe the text to state that the global explanation is only valid for the fraction of data where the tree and XGBoost agree. They must calculate and display an explicit "fidelity metric" or track how the explanation's utility breaks down in cases of model disagreement.

2. In Figure 22, the comparison against an XGBoost baseline trained on 10k samples must be removed or fixed. The authors must train an XGBoost baseline on the identical 50k dataset split to ensure an equitable comparison of model accuracy.

3. Given that the MIP gap remains at 60%, the authors need to add an ablation study detailing how much of the final performance is inherited directly from the scikit-learn CART warmstart versus the optimization steps taken by Gurobi.

4. Equation 1 on Page 3 is poorly formatted with incomplete OCR-like fragments ($\Sigma y = C_i$, $X_1$, etc.). This must be replaced with an exact, mathematically sound notation for $AL(T)$.

5. Given the high computational footprint (15,500 core-hours), the authors should discuss or implement a brief test using recent specialized dynamic programming/branch-and-bound tree solvers mentioned in their related work to show if the 8-hour bottleneck can be scaled down.

6. Provide a small example mapping where (in which leaves) the localized XGBoost models tend to disagree most with the shallow tree's boundaries.

---

> ### Author Response · Authors · 2026-06-05
> **Rebuttal 1/2**
>
> We thank the reviewer for their extensive review and for their appreciation of our work. While it is true that using MIO is more computationally heavy, we show that it can outperform existing methods with modest resources after 1 hour (see Figure 8). While the extension with XGBoost indeed hinders interpretability, it is presented as an optional step when aiming for model performance, and was chosen to compare with the top-performing model. We respectfully disagree with the assessment that our evaluation is set up to favor our method. We compare with the models as they were proposed and as they are being used.
>
> We would like to flag a correction to the agreement rate figures reported in the original draft. The corrected disagreement rate is approximately 19%, roughly half the originally reported 37.7%. We have updated the draft accordingly. This reduces the reviewer’s concern regarding the reliability of the shallow tree as an explanation in the extended setting.
>
> Going point-by-point, regarding the three main concerns:
>
> 1) It is important to note that the model extension is an optional step, aimed at improving the predictive performance. The fact that a portion of the data is not well explained is common in partially interpretable models, as noted in the limitations paragraph. Our claim of improved usefulness rests on the improvement in leaf accuracy of the tree itself, compared to a tree generated by a different method prior to tree extension. In line with the reviewers’ broader impact concerns, we have added a paragraph to the limitations, explicitly noting the difference in interpretability between the base and extended models.
> 2) While MIO has global optimality capabilities, we do not claim it finds globally optimal solutions to our formulation with the given resources. The MIP gap indeed confirms that solutions are not proven optimal. However, we empirically show that it dominates the solutions found by OCT, which was warmstarted with the same initial solutions, or solutions found by CART, which are similar to those used for warmstarting. We have also added a new section to the appendix (A.10), where we compare the models created by our method with the CART solutions used to warmstart them. The results are similar to the results of tuned CART models. Finally, Figure 16 shows the non-warmstarted variant(s) of our method outperforming other approaches, suggesting that CART is a useful initial solution, but not necessary for the success of our approach.
> 3) We chose to show the warmstarted OCT results because this approach was shown to perform best in the original work (Bertsimas & Dunn, 2017). Our method is also warm-started (with the same initial solution), so this comparison seemed most fair. The fact that non-warmstarted OCT performs better in terms of leaf accuracy was indeed surprising, though it still performs worse than the proposed method. It may plausibly be due to OCT creating models with limited complexity (e.g., depth-1 trees), given the poor scores in overall model accuracy. Such models could achieve a more stable (i.e. generalizable) leaf accuracy, since fewer leaves would contain larger portions of data.

---

> > ### Author Response · Authors · 2026-06-05
> > **Rebuttal 2/2**
> >
> > In this second part, we address the requested changes:
> >
> > 1) We have rephrased portions of the text that might be misleading in terms of the applicability of the tree as a global explanation. We strictly distinguish the tree itself as a potential global explanation and the extended model as merely a partially explainable model. We have added a closer comparison of the agreement rates across datasets and a further discussion in Appendix A.11.
> > 2) Figure 22 contained results of other methods using 10k samples solely to better compare the effect of increasing the number of samples. As per the reviewer’s request, we have removed the figure, opting to only describe the effect in the corresponding section.
> > 3) The comparison of CART trees and trees created by the proposed method was already in the paper. In terms of performance, we compare to the non-warmstarted variant in Figure 16. In Figure 10, we compare the number of leaves in trees generated by different methods, showing that the trees are also structurally different. Finally, we have added another section to the appendix, studying the similarity of the final models compared to the warmstarting trees.
> > 4) It is unclear to us what is unsound about Equation 1\. All symbols are explained in the paragraph immediately following the equation. $X\_1$ is not present in the equation, only $X\_l$, which represents the set of samples assigned to the leaf $l$, as explained in the text. The double brackets surrounding $y \= C\_l$ represent the Iverson bracket, or an indicator function, as explained in the text. The equation is typeset in a standard way, using LaTeX. We will be happy to correct any mistakes if they are clarified.
> > 5) In Figure 8, we show that results even after 1 hour show a noticeable improvement over CART, suggesting that the full 8-hour time limit may not always be necessary. Because we modify the objective function of the tree learning, modifying other methods suggested in the related works to consider leaf accuracy instead would be non-trivial. It is certainly beyond the scope of what can be done given the time constraints of the rebuttal and is left for future work.
> > 6) We have introduced a new case study in Appendix A.11.1, showcasing the final model and the tree used to warmstart the solver, with the information about agreement rates included. A majority of leaves agree on all assigned classifications on the test set, and only a few have agreement below 80%.

---

> > ### Comment · Reviewer_44nq · 2026-06-27
> > **The disagreement rate is approximately 19%, it is still high and should not be unconsidered.**
> >
> > I appreciate your approach to improving the paper.
> > However, I think that a disagreement rate of 19%, or predictive multiplicity as it is more commonly known in the literature, is high and cannot be ignored.
> > I suggest acknowledging this as a potential shortcoming will encourage readers and users to exercise caution when using your methodology.

---

> > > ### Author Response · Authors · 2026-07-01
> > > **The disagreement rate is reported and its implications discussed.**
> > >
> > > We thank the reviewer for their comment and agree that this disagreement rate is non-trivial and should not be ignored. We report it in the limitations section and devote a paragraph to cautioning users: we recommend that they investigate and report this rate and avoid treating the shallow tree as a full explanation of the hybrid model.
> > >
> > > In Appendix A.11, we provide more details, including rates for each dataset (Figure 17) and a case study comparing two models trained on a single data split (Figure 18).
> > >
> > > We are happy to further improve our handling of this topic and welcome any concrete suggestions.

---

### Review · Reviewer_JHGe · 2026-05-28

**Summary Of Contributions:**

The manuscript presents a learning algorithm for decision trees. The algorithm maximizes the accuracy of the least accurate leaf of the tree, thus improving the reliability of the least reliable explanation the tree can generate. The model is based on a MIP formulation that extends the one of Optimal Trees.
Moreover, the manuscript presents an hybrid modeling in which additional trees are learned on each leaf to improve expressivity.

**Additional Comments:**

# Typos
- Fig. 3 the legend headers
- "The Proposed method", page 8

**Audience:**

Yes

**Audience Explanation:**

(Optimal) Decision Trees are central in interpretable modeling.

**Claims And Evidence:**

No

**Claims Explanation:**

- Some details are unclear:
	- It is not clear if the experiments in Figure 3 include model extensions or not. The paragraph ("Our *partially* interpretable model [..]") seems to suggest so, which would make only xgboost the suitable competitor
	- Is the mean leaf accuracy change computed on train, or on test? I.e., does the problem reside in learning or generalization?
	- Normalized leaf accuracy appears to measure leaf performance imbalance ($\dfrac{leaf \text{ } accuracy}{\max {leaf \text { } accuracy}}$), rather than min leaf accuracy: why was it chosen?
- There are some questions I think deserve research, and that should be addressed:
	- What is the impact of the reduction on the accuracy of the min. accuracy leaf?
	- What is the model performance difference of the base tree VS tree + tree extension?
	- How do the tree extension compare w/ similar models, e.g. Model Trees? (Using Model Trees for Classification. Frank et al.)
	- How does the model's complexity compare w/ CART?

**Requested Changes:**

For the sake of completeness, I also list some strong/weak points of the manuscript

# Strong points
1. The manuscript tackles an understudied yet practical problem
2. Simple formulation

# Weak points
1. Contribution is rather minimal (additional constraints to an existing formulation)
2. Experiments miss some minor research questions
3. Some important analyses are not tackled in much depth in the manuscript

# Observations
- I find the warm start for the MIO seems a bit counterintuitive: to improve on CART models, we first train a CART model, then optimize with MIP, then compare against CART? The setup seems to unfairly punish the competitor. This point could be better addressed in the manuscript
- (strengthen) Many relevant questions are left to the appendix, I think they'd be better presented in the core manuscript, e.g.,
	- optimality gap (A.3)
	- time required to outperform CART (A, 8.c)
- Some details are unclear:
	- (strengthen) It is not clear if the experiments in Figure 3 include model extensions or not. The paragraph ("Our *partially* interpretable model [..]") seems to suggest so, which would make only xgboost the suitable competitor
	- (strengthen) Is the mean leaf accuracy change computed on train, or on test? I.e., does the problem reside in learning or generalization?
	- (strengthen) Normalized leaf accuracy appears to measure leaf performance imbalance ($\dfrac{leaf \text{ } accuracy}{\max {leaf \text { } accuracy}}$), rather than min leaf accuracy: why was it chosen?
- (critical) There are some questions I think deserve research, and that should be addressed:
	- What is the impact of the reduction on the accuracy of the min. accuracy leaf?
	- What is the model performance difference of the base tree VS tree + tree extension?
	- How do the tree extension compare w/ similar models, e.g. Model Trees? (Using Model Trees for Classification. Frank et al.)
	- How does the model's complexity compare w/ CART?

Overall, I feel the manuscript loses a bit of focus by tackling both min. leaf accuracy and extension (particularly when it does not compare with other extension models). I think the two are rather different research questions w/ enough depth (pun not intended) to be researched separately

---

> ### Author Response · Authors · 2026-06-05
> **Rebuttal**
>
> We thank the reviewer for their thorough review and for recognizing the tackled problem as practical and understudied. We understand the reviewer’s overall concern that we are tackling two separate questions: leaf accuracy and tree extension. The purpose of the extension is primarily to showcase an option for improving overall performance, which is why we do not compare it to other hybrid tree architectures. The paper thus tackles only the question of leaf accuracy in depth. We have reworded the paper narrative to make this more explicit.
>
> Going through the observations point-by-point:
>
> 1) Warmstarting the MIO solver is a standard practice that improves performance by providing an initial feasible (ideally also high-quality) solution, thus introducing a bound on the objective that can be used in the branch-and-bound algorithm. Our approach is also in line with the original OCT formulation, which also used CART trees to warmstart the solver (and compared to CART afterward). Notably, unlike the CART we compare to, the warm-starting CART solutions do not undergo hyperparameter tuning. We take the default solution, limiting only the depth and the minimal number of samples in leaves to ensure feasibility. Finally, the proposed method outperforms CART even without warmstarting, see Figure 16\.
> 2) We have expanded Section 4 of the main body, adding a paragraph on scalability, with the requested topics discussed.
> 3) Unclear details:
>    1) Experiments represented by the diamonds in Figure 3 include model extensions. As per the legend, we refer to extended trees as hybrid trees, in line with the literature. Leaf accuracy is always computed on predictions of the shallow trees, as defined in Eq. 1\.
>    2) All presented results were computed on the test set, unless explicitly specified otherwise, or unless it is clear from context (e.g., MIP Gap is computed on the training set). Because leaf accuracy takes the minimum over the leaves, a single poorly fitted leaf can degrade the entire tree’s performance. Leaf accuracy is thus often notably worse on test data compared to the training set, oftentimes due to the low number of samples in leaves, which leads to poor generalization. See Figure 20 for a comparison to a formulation without a lower bound on the sample count per leaf, showing that without this limit, our method's performance collapses due to poor generalization.
>    3) The normalized leaf accuracy is normalized across methods over a single split of a single dataset. It is (leaf accuracy of a tree)/(leaf accuracy of the tree with maximal leaf accuracy). Its purpose is to reduce variance and compare methods more directly in Figure 4, since Figure 3 shows large variance due to aggregation across multiple datasets with different data complexities. We understand the confusion; the relevant section has been reworded for clarity.
> 4) Critical points
>    1) The effect of reduction is shown in Figure 9\. It shows a notable improvement in mean leaf accuracy.
>    2) It is unclear to us what is meant by this question. We show the difference in performance of the base tree and the extended (i.e., hybrid) tree in all relevant figures (Fig. 2,3,5,8,11-16,19-21,23). The squares represent the performance of base tree models, and the diamonds represent hybrid trees, i.e., those with leaves extended.
>    3) In the terminology we use, model trees are hybrid trees, where leaves are extended with linear regression models. Our approach extends leaves with XGBoost models to better compare with the baseline model on the used benchmark. Since the aim of the paper is to study leaf accuracy in decision trees, rather than hybrid models, we compare only to hybrid trees extended in the same way, with base trees created by CART or OCT methods. The results suggest that base trees optimized for leaf accuracy might perform better when extended, though we do not investigate this question in depth, since we considered it out of scope. We agree with the reviewer that studying hybrid trees in depth is a separate research question. This distinction was clarified throughout the paper.
>    4) The comparison of model complexity between the CART and proposed model is in Figure 10, which shows the distributions of the number of leaves in trees generated by each method. Trees generated by the proposed method have fewer leaves, suggesting that low-accuracy leaves are pruned away (similar to the example in Figure 1).

---

### Comment · Action_Editor_yrM4 · 2026-06-07
**Reviewer-author discussion**

Dear reviewers, the authors have now responded to your reviews and it is time to engage with them in any subsequent discussion.

I also advise you to check not only your part of the review and responses but also the comments of the other reviewers and the related discussion. Such crosscheck may give you a better feeling for the paper, help in solidifying your own views, and may be particularly useful for reaching your final recommendation.

Thanks,
Your AE